# The Stroop effect involves an excitatory–inhibitory fronto-cerebellar loop

Moe Okayasu [1], Tensei Inukai [1], Daiki Tanaka [1], Kaho Tsumura [1], Reiko Shintaki [1], Masaki Takeda[2], Kiyoshi Nakahara [2] & Koji Jimura [1,2,3] ✉

The Stroop effect is a classical, well-known behavioral phenomenon in humans that refers to robust interference between language and color information. It remains unclear, however, when the interference occurs and how it is resolved in the brain. Here we show that the Stroop effect occurs during perception of color–word stimuli and involves a cross-hemispheric, excitatory–inhibitory loop functionally connecting the lateral prefrontal cortex and cerebellum. Participants performed a Stroop task and a non-verbal control task (which we term the Swimmy task), and made a response vocally or manually. The Stroop effect involved the lateral prefrontal cortex in the left hemisphere and the cerebellum in the right hemisphere, independently of the response type; such lateralization was absent during the Swimmy task, however. Moreover, the prefrontal cortex amplified cerebellar activity, whereas the cerebellum suppressed prefrontal activity. This fronto–cerebellar loop may implement language and cognitive systems that enable goal-directed behavior during perceptual conflicts.

The Stroop effect is widely acknowledged as a robust and intriguing behavioral phenomenon referring to a prolonged reaction time when naming the font color of a printed word if this color differs from that represented by the word's meaning[1–4] (Fig. 1a). It is thought to be attributable to interference between language and color information, and resolution of the interference requires high cognitive control.

The Stroop effect is possibly unique to humans since it relates to language functions involved in reasoning, problem-solving, and other elaborated processing operations that characterize flexible human behavior[5–10]. Thus, one notable signature of the Stroop effect is that these language functions persistently interfere with color information[2,11–15]. Despite the importance and long history of this effect, it remains unclear when it occurs and how it is resolved.

The Stroop effect was originally reported based on vocal responses[1,13,16], which involve two stages of language processing: perception of a word stimulus and generation of a vocal response (Fig. 1a). It is well known that distinct regions of the brain are responsible for these two stages[17,18]. Behavioral studies have also suggested that the vocal and manual response involve distinct processing during the resolution of

the Stroop effect[15,19,20]. These observations suggest that response generation and its underlying neural mechanisms play an important role in the Stroop effect. However, previous neuroimaging studies of the vocal response[21–24], manual response[12,14,23,25–27], and covert response[23,28,29] have consistently suggested that the Stroop effect is associated with the anterior cingulate, medial prefrontal, and bilateral prefrontal cortices. Nonetheless, bilateral and medial prefrontal involvement is inconsistent with the traditional view that the human language system is predominantly implicated in the left hemisphere[18,30].

While Stroop tasks induce interference of verbal information processing, cognitive interference occurs in the absence of verbal information as demonstrated by flanker tasks. In a common flanker task, verbal information is not involved in either stimulus perception or response generation[31], whereas other flanker-type tasks use a verbal stimulus[32,33]. Interestingly, as is the case in Stroop tasks, the interference caused by flanker tasks involves the anterior cingulate, bilateral prefrontal, and parietal cortices[32–38]. This raises a question about whether the resolution of Stroop- and flanker-type non-verbal interference shares core brain mechanisms. If this were the case, such

[1]Department of Biosciences and Informatics, Keio University, Yokohama, Japan. [2]Research Center for Brain Communication, Kochi University of Technology, Kami, Japan. [3]Department of Informatics, Gunma University, Maebashi, Japan. ✉e-mail: jimura@gunma-u.ac.jp

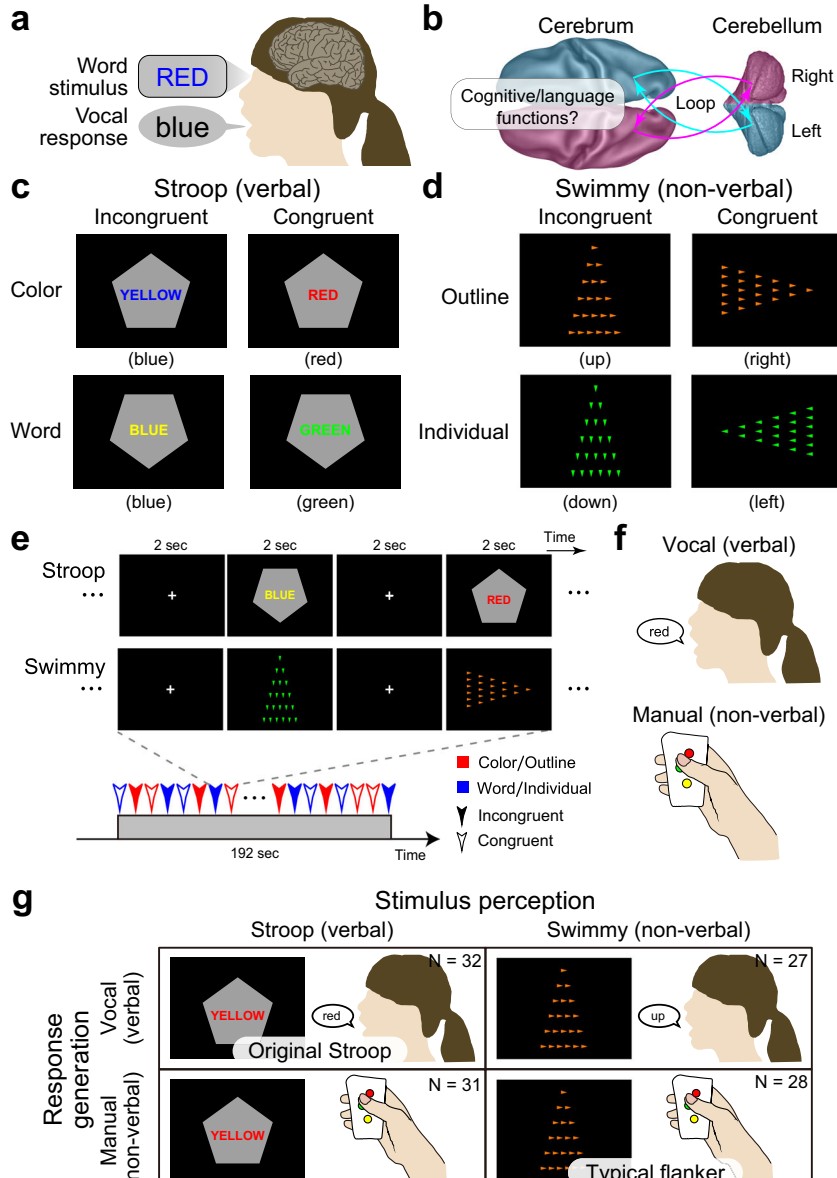

**Fig. 1 | Experimental design to examine the role of language processing in the resolution of the Stroop effect. a** Stroop effect. It takes longer to name the color of a colored word when the word is colored in an inconsistent color (e.g., the word "RED" in blue-color font). **b** A schematic illustration of cross-hemispheric cerebro-cerebellar loops. **c** Stroop tasks. Participants made a judgment regarding the color (top) or word (bottom) of a colored word (Stroop task). The judgment was indicated by the orientation of the vertex of a pentagon (up: color; down: word). The word color and colored word were inconsistent (incongruent; left) or consistent (congruent; right). **d** Control (Swimmy) tasks. Another set of participants made a judgment regarding the orientation of the vertex of a large outline of a triangle (top) or of small individual (bottom) triangles (Swimmy task). The task dimension as indicated by the color of the triangles (orange: outline; green: individual). The orientation of the outline and individual triangles was inconsistent (incongruent; left) or consistent (congruent; right). **e** Behavioral procedures. In both the Stroop and Swimmy tasks, four trial conditions (two levels of tasks and two levels of congruency) were presented pseudorandomly. **f** Participants responded vocally (left) or manually (right). **g** The experimental conditions configured a 2 × 2 factorial design.

mechanisms would play a generic role in the resolution of cognitive interference beyond the verbality of the stimulus and response[23]; if not, the resolution would depend on the verbality, suggesting that language processing is specifically involved in the Stroop effect[33].

It is widely accepted that the brain regions responsible for cognitive and language functions are associative neocortical regions distributed in the cerebrum[17,18,30,39]. Increasing evidence suggests that the cerebellum also plays important roles in language and cognitive control[40–49]. This cerebellar involvement is associated with dorsal regions in the lateral hemispheres (crus I/II, lobules VI/VIIb)[41,45,46], and is independent of sensorimotor functions implicated in rostral and caudal ventromedial regions (lobules I-VI, VIIIa/b)[41,50,51]. These cerebellar

regions constitute a cortico-cerebellar loop between cerebral cortical regions in the contra-lateral hemisphere[41,44,50,51] (Fig. 1b). Importantly, damage to the dorsolateral cerebellar regions impairs language functions[41,45,52], and a classical neuroimaging study reported that language processing without vocal response involves a cerebellar region in the right hemisphere[52]. This collective evidence suggests that a cross-hemispheric cerebro-cerebellar loop is involved in cognitive and language functions (Fig. 1b). Accordingly, we asked whether the cerebellum plays an important role during the resolution of the Stroop effect involving cognitive and language processing.

In the current study, we examined the role of language processing in the Stroop effect by manipulating the verbality of stimulus

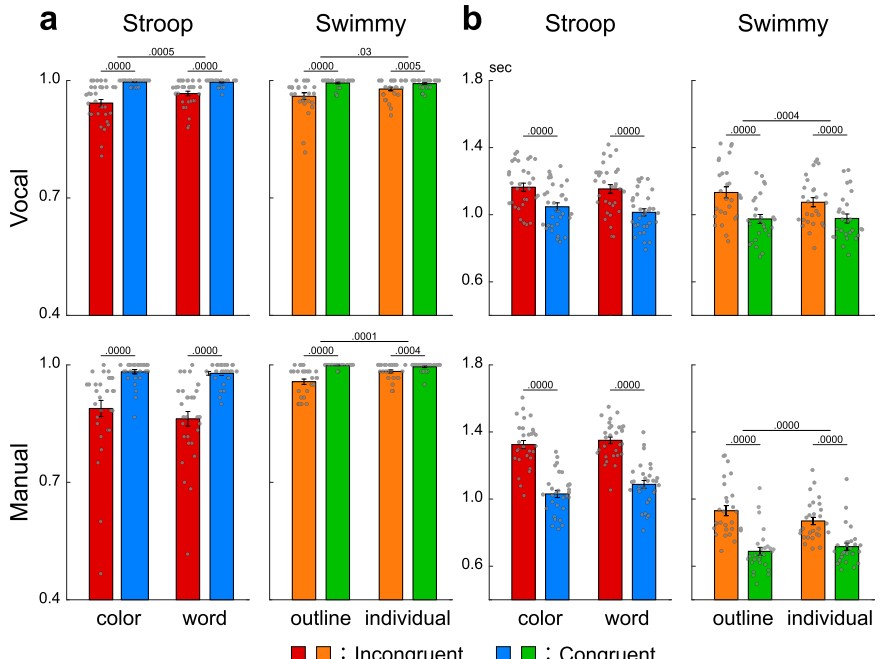

**Fig. 2 | Behavioral results. a** Accuracy in the Stroop (left) and Swimmy (right) tasks with vocal (top) and manual (bottom) responses (vocal Stroop: $N = 32$; manual Stroop: $N = 31$; vocal Swimmy: $N = 27$; manual Swimmy: $N = 28$; independent participants). The vertical and horizontal axes indicate accuracy and conditions (task and congruency), respectively. Rectangular and error bars indicate means and standard errors of the mean across participants, respectively, with individual data overlaid on each rectangular bar. Statistical $p$ values are shown on the top (paired $t$ tests, two-tailed, uncorrected). Red: incongruent Stroop; blue: congruent Stroop; orange: incongruent Swimmy; green: congruent Swimmy. **b** Reaction times. The vertical and horizontal axes indicate reaction times and conditions, respectively. Statistical procedures and formats are similar to those in **a**. The sample size is identical to that in **a**.

perception and response generation involved in the interference resolution. Specifically, we conducted a set of functional MRI experiments where humans performed a Stroop task or a non-verbal control task (which we term the Swimmy task) using vocal or manual responses (Fig. 1c–g). The Swimmy task served as a control condition in which the stimulus did not involve language information, similar to a flanker task combining the Simon and local–global tasks[53,54]. Then, we comprehensively explored language processing during interference in the Stroop and Swimmy tasks (Fig. 1g), and examined directional effective connectivity between responsible brain regions to identify task-related signal flows between the regions.

## Results
### Behavioral tasks
A group of human participants ($N = 63$) performed a Stroop task. In each trial, a colored word was presented on a gray, pentagon-shaped background, and participants were required to judge the color or word of the stimulus, depending on the orientation of the pentagon vertex (Fig. 1c). The color and word were either incongruent or congruent.

Another group of participants ($N = 55$) performed another task that served as a control of the Stroop task in terms of stimulus verbality, in which the stimulus did not involve verbal information (Fig. 1d). We named this task the Swimmy task after an old picture book in which small fish were arranged in a pattern that resembled a large fish[55]. In each trial, a set of small, individual triangles was presented on the screen to form the outline of a larger triangle. Participants judged the orientation of the outline or individual triangles (in each case based on the vertex with the smallest angle), depending on the color of the individual triangles. The orientations of the outline and individual triangles were either incongruent or congruent.

In both the Stroop and Swimmy tasks, the task procedures were matched except for the nature of the visual stimulus (Fig. 1e), and participants responded either vocally (Stroop: $N = 32$; Swimmy: 27) or manually (Stroop: $N = 31$; Swimmy: 28) (Fig. 1f). Thus, the current

experimental design consisted of two levels of stimulus modality [verbal (Stroop), non-verbal (Swimmy)] and two levels of response modality (vocal, manual), entailing a 2 × 2 factorial design (Fig. 1g). In the vocal conditions, responses were recorded through an MRI-compatible noise-reduction microphone (Supplementary Fig. 1).

### Behavioral performance
In the Stroop task, accuracy was lower in incongruent than congruent trials in both the vocal and manual conditions [vocal: $F(1, 31) = 44.0$, $P < 0.001$; manual: $F(1, 30) = 48.5$, $P < 0.001$; Fig. 2a left]. In the vocal Stroop task, the interaction effect of congruency (incongruent and congruent) and task (color and word) was also significant [$F(1, 31) = 9.0$, $P < 0.01$], but such interaction was absent in the manual condition [$F(1, 30) = 1.5$, $P = 0.23$]. Accuracy was lower in the manual than the vocal condition [$F(1, 61) = 20.0$, $P < 0.001$]. The interaction effect of response modality (vocal and manual) and congruency was significant [$F(1, 61) = 15.6$, $P < 0.001$].

In the Swimmy task, accuracy was lower in the incongruent than the congruent trials in both the vocal and manual conditions, similar to the Stroop task [vocal: $F(1, 26) = 18.1$, $P < 0.001$; manual: $F(1, 27) = 40.6$, $P < 0.001$; Fig. 2a right]. In the vocal Swimmy task, the interaction effect of congruency (incongruent and congruent) and task (outline and individual) was also significant [vocal: $F(1, 26) = 5.5$, $P < 0.05$; manual: $F(1, 27) = 20.0$, $P < 0.001$], indicating that the interference effect was greater in the outline task. There was no significant difference in accuracy between the vocal and manual conditions [$F(1, 53) = 0.45$, $P = 0.51$]. The interaction effect of response modality and congruency was not significant [$F(1, 53) = 0.17$, $P = 0.68$].

In the vocal conditions, reaction times were calculated as the latency from stimulus onset to vocal response onset (Supplementary Fig. 1). In the Stroop task, reaction times were longer in the incongruent than congruent trials in both the vocal and manual conditions [vocal: $F(1, 31) = 123.5$, $P < 0.001$; manual: $F(1, 30) = 149.3$, $P < 0.001$; Fig. 2b left]. The interaction effect of

congruency and task (color and word) was insignificant in both response modalities [vocal: F(1, 31) = 3.6, P = 0.07; manual: F(1, 30) = 4.1, P = 0.05]. Reaction times were slower in the manual than vocal conditions [F(1, 61) = 13.3, P < 0.01]. The interaction effect of response modality and congruency was significant [F(1, 61) = 35.3, P < 0.001].

In this study, reaction times were longer than in the classical Stroop studies[11] because of the pseudorandomized event-related fMRI design in which baseline cognitive demand was high. However, the interference effect (i.e., incongruent vs. congruent) was 117 ± 67 (mean ± SD) ms in the color task and 140 ± 80 ms in the word task in the vocal conditions, and 294 ± 145 ms in the color task and 263 ± 123 ms in the word task in the manual condition, values that were comparable to those in previous studies[15,19,20].

In the Swimmy task, reaction times were longer in the incongruent than congruent trials in both the vocal and manual conditions, similar to the Stroop task [vocal: F(1, 26) = 111.0, P < 0.001; manual: F(1, 27) = 196.4, P < 0.001; Fig. 2b right]. The interaction effect of congruency and task was significant in both response modalities [vocal: F(1, 26) = 16.8, P < 0.001; manual: F(1, 27) = 62.1, P < 0.01], which is attributable to greater interference effect in the outline task. Reaction times were longer in the vocal than manual conditions [F(1, 53) = 45.2, P < 0.001]. The interaction effect of response modality and congruency was significant [F(1, 53) = 14.3, P < 0.001].

These collective behavioral results demonstrated that cognitive interference was successfully imposed during the incongruent trials in both the Stroop and Swimmy tasks using vocal and manual responses.

## The Stroop effect involves cross-hemispheric cerebro-cerebellar mechanisms

In an imaging analysis, we first explored brain regions associated with the interference effect (i.e., incongruent vs. congruent trials) during the Stroop task (vocal and manual conditions collapsed; N = 63). A strong interference effect was observed in the lateral prefrontal cortex (lPFC), posterior parietal cortex (PPC), and occipitotemporal cortex (OTC) (Fig. 3a and Supplementary Table 1; the vocal and manual conditions are collapsed). Interestingly, these prominent activations in cortical regions were observed mainly in the left hemisphere. The cerebellum also showed a strong interference effect, specifically in the crus I/lobule VI (a dorsal and caudal region) and the crus II/lobule VIIb (a dorsal and rostral region) of the cerebellar hemisphere. Notably, this cerebellar involvement was observed mainly in the right hemisphere, which contrasts with the left lateralization in the cortical regions. Involvement of the cortical and cerebellar regions was consistently observed in the vocal and manual conditions when these were analyzed separately (Supplementary Fig. 2). The medial prefrontal cortex (mPFC) and anterior cingulate cortex (ACC) also showed interference effects, which is consistent with the fact that these regions are considered to play a key role in the resolution of the Stroop interference[12,16,23–25,29,56–63]. The activations look weaker compared to those in the left lateral cerebral and right cerebellar regions, however.

Prior behavioral studies have demonstrated asymmetry of interference effects in the color-naming task (i.e., Stroop effect) and word-naming task (i.e., reverse-Stroop effect)[64–67]. In particular, the interference effects were greater in the color-naming task than in the word-naming task, suggesting stronger cognitive demand to attend to color. Indeed, our behavioral analysis of accuracy in the vocal condition showed a significant interaction effect of congruency and task (color and word), an asymmetric interference pattern that has been observed in reaction times[11]. To test whether the variability in behavioral interference effects is reflected in brain activity, we explored brain regions associated with the interference effect for the color and word tasks separately. In both the color and word tasks, a strong interference effect was observed in the lPFC, PPC, and OTC, all mainly in the left

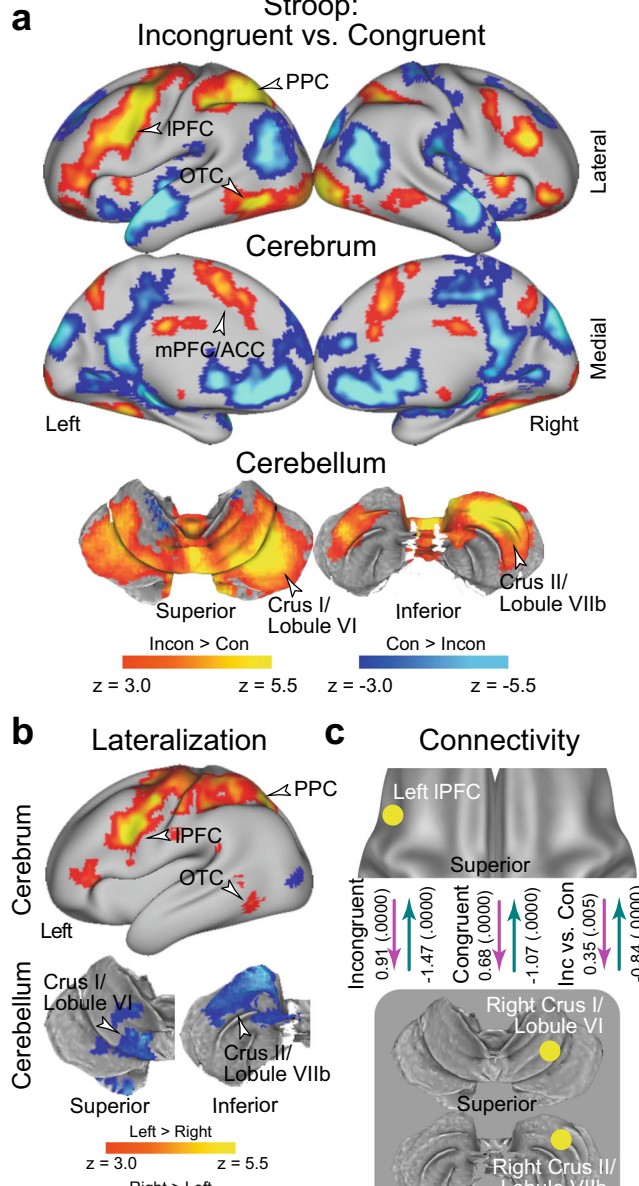

**Fig. 3 | The Stroop effect involves a lateralized fronto-cerebellar loop during stimulus perception. a** Statistical activation maps for signal increase and decrease in the contrast between incongruent and congruent trials in the Stroop task (P < 0.05, FWE-corrected across the whole-brain based on non-parametric permutation tests). The vocal and manual conditions were collapsed. Maps are overlaid onto a 3D surface of the brain. Hot and cool colors indicate signal increase and decrease in the incongruent trials, respectively. Arrowheads indicate anatomical locations of major activations. lPFC: lateral prefrontal cortex; PPC: posterior parietal cortex; OTC: occipitotemporal cortex; mPFC: medial prefrontal cortex; ACC: anterior cingulate cortex. **b** Statistical maps of brain regions showing differential Stroop effect activation (**a**) between the left and right hemispheres. Hot and cool colors indicate greater activity in the left and right hemispheres, respectively. **c** Task-related effective connectivity analysis between lPFC and cerebellar regions based on dynamic causal modeling. The values indicate estimates of the connectivity and their p-values calculated based on posterior probability density (one-tailed, uncorrected) are shown in parentheses next. The arrow directions indicate task-related effective connectivity. The magenta and green arrows indicate positive and negative effects, respectively.

hemisphere, and also in right cerebellar regions; overall, the activation maps look similar for both tasks (Supplementary Fig. 3a/b). When comparing the neural interference effect between the color and word tasks, differential activity was almost absent (Supplementary Fig. 3c). Our results suggest that the Stroop and reverse-Stroop effects involve common neural mechanisms, whereas a previous study reported differential effects in cortical regions[67].

We then asked whether the neocortical and cerebellar regions are lateralized. To this end, we contrasted the activation maps for the interference effect between the left and right hemispheres on a voxel-by-voxel basis, and then explored the brain regions showing a differential interference effect between hemispheres[68] (Supplementary Fig. 4; see also Materials and Methods). Greater activity in the left hemisphere was observed in the lPFC, PPC, and OTC. In the cerebellum, on the other hand, the crus I/lobule VI and crus II/lobule VIIb showed greater activity in the right hemisphere (Fig. 3b). These results clearly demonstrate cross-hemispheric involvement in the neocortex and cerebellum.

Given the cross-hemispheric, lateralized cerebro-cerebellar involvement, we next asked how the cortical and cerebellar regions interacted during the resolution of Stroop interference. To address this issue, we performed an interregional effective connectivity analysis based on dynamic causal modeling (DCM), which makes it possible to examine the directionality of task-related functional connectivity based on the state-space model (see Materials and Methods)[68–74]. The regions of interest (ROIs) were defined as the left lPFC and the right cerebellar regions (CER; crus I/lobule VI and crus II/lobule VIIb), all of which showed robust activation (Fig. 3a) and lateralization (Fig. 3b). To avoid circular analysis[75], the ROIs were defined independently of the tested data. Specifically, for the analysis of the vocal condition, the lPFC and CER ROIs were defined as the regions showing the interference effect in the manual condition (Supplementary Fig. 2b), and vice versa (Supplementary Fig. 2a).

In both the incongruent and congruent trials, the connectivity was excitatory from the lPFC to the CER ($P < 0.001$), but inhibitory from the CER to the lPFC ($P < 0.001$) (Fig. 3c). Interestingly, the excitatory connectivity from the lPFC to the CER and the inhibitory connectivity from the CER to the lPFC were strengthened in the incongruent trials relative to the congruent trial ($P < 0.001$; Fig. 3c). The original results were confirmed when the lPFC ROI was defined based on a meta-analysis map of cognitive control[62] (see Methods) (Supplementary Fig. 5).

When the crus I/lobule VI and crus II/lobule VIIb were analyzed separately, consistent results were obtained, suggesting that these two cerebellar regions implement homologous functions in the Stroop effect (Supplementary Fig. 6a/b). Moreover, these connectivity results were observed in the vocal and manual conditions consistently (Supplementary Fig. 6c–h), suggesting that the functional connectivity between the lPFC and the CER was independent of response modality. On the other hand, the color task showed stronger excitatory connectivity and weaker inhibitory connectivity relative to the word task (Supplementary Fig. 7).

### The Swimmy effect involves bilateral cerebro-cerebellar mechanisms

In the Swimmy task, a strong interference effect (incongruent vs. congruent trials) was observed bilaterally in the lPFC, PPC, and OTC. In the cerebellum, the crus I/lobule VI (a dorsal and caudal region) and the crus II/lobule VIIb (a dorsal and rostral region) also showed strong bilateral interference effects (Fig. 4a and Supplementary Table 2). The involvement of these cortical and cerebellar regions was predominant in both the vocal and manual conditions when these were analyzed separately (Supplementary Fig. 8). These results suggest that the resolution of the Swimmy effect involves bilateral cortical and cerebellar regions, independently of response

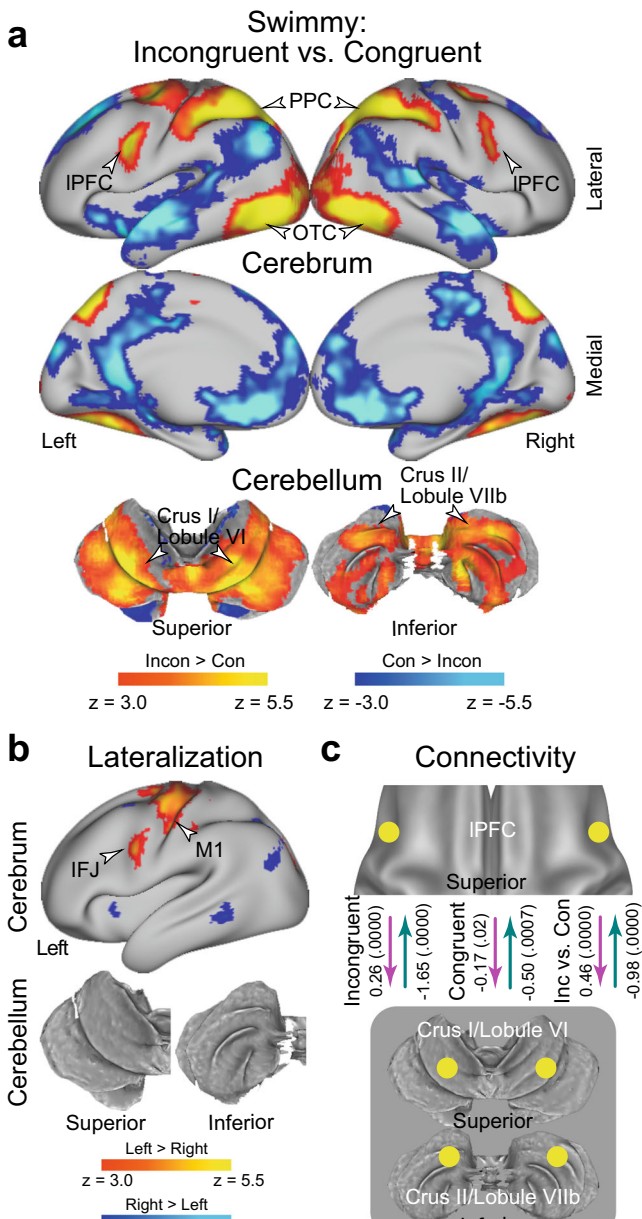

**Fig. 4 | Non-verbal Swimmy effect involves a bilateral fronto-cerebellar loop during stimulus perception. a** Statistical activation maps for signal increase and decrease in the contrast between incongruent and congruent trials in the Swimmy task. The vocal and manual conditions were collapsed. **b** Statistical maps of brain regions showing differential Swimmy effect activation (**a**) between the left and right hemispheres. IFJ inferior frontal junction. **c** Effective connectivity analysis between the lPFC and cerebellar regions based on dynamic causal modeling. The dashed arrows indicate insignificant connectivity. Statistical procedures and formats are similar to those in Fig. 3.

modality. This response modality-independent involvement was consistent with that observed in the Stroop task. The outline and individual tasks involved similar cortical and cerebellar regions (Supplementary Fig. 9a, b), but the interference effect differed between the two tasks in multiple neocortical and cerebellar regions (Supplementary Fig. 9c), possibly reflecting distinctive attention to global and local shapes[76].

We then explored the laterality of activity during the Swimmy interference. The inferior frontal junction (IFJ) and the primary motor cortex showed stronger activity in the left hemisphere (Fig. 4b);

however, compared to the Stroop task (Fig. 3b), the lateralization was weaker across the neocortical regions. It should be noted that the IFJ region is located in the posterior bank of the precentral sulcus (Fig. 3b), and is spatially separated from the lPFC region located in the inferior frontal sulcus. In the cerebellum, there was no lateralized activity. These results suggest that relative to the Stroop effect, the Swimmy effect involves neocortical and cerebellar areas more bilaterally.

We next performed DCM analysis of bilateral lPFC and CER ROIs. In the incongruent trials, the connectivity was excitatory from the lPFC to the CER, and inhibitory from the CER to the lPFC (Fig. 4c) ($P < 0.001$). However, connectivity was weak in the congruent trials ($P$s > 0.07). The excitatory connectivity from the lPFC to the CER and the inhibitory connectivity from the CER to the lPFC was greater in the incongruent trials than in the congruent trials ($P$s < 0.001; Fig. 4c). The original results were confirmed when the lPFC ROI was defined based on a meta-analysis map of cognitive control[62] (see Methods) (Supplementary Fig. 10), similarly to the DCM analysis for the Stroop effect (Supplementary Fig. 5).

Again, these results are consistently observed in both the vocal and manual conditions when the two conditions were analyzed separately (Supplementary Fig. 11a/b). DCM analysis of the lPFC and CER in unilateral and cross-hemispheric ROIs also exhibited excitatory and inhibitory connectivity (Supplementary Fig. 11c–f). This suggests that in terms of functional involvement and effective connectivity, the left and right hemispheres implement homogeneous functionality during the Swimmy task, in contrast to the Stroop task. Connectivity strength was not robustly differed between the outline and individual tasks (Supplementary Fig. 12).

### Fronto-cerebellar involvement is independent of response modality
To examine whether response modality (vocal vs. manual) is critical for the interference effect, we directly compared brain activity between the vocal and manual conditions (Supplementary Fig. 13). When the incongruent and congruent trials were collapsed, the vocal condition resulted in greater brain activity in the primary auditory cortex, ventrolateral parts of the primary motor cortex, and dorsal parts of cerebellar lobules V/VIIIa/VIIb, all of which are implicated in vocalization (arrowheads labeled "vocalization" in Supplementary Fig. 13a, b). On the other hand, the manual condition led to greater activity in the dorsomedial parts of the primary motor cortex and somatosensory cortex, and the ventral parts of cerebellar lobules V/VI/VIIIa/VIIIb, all of which which are implicated in right-hand movement (arrowheads labeled "vocalization" in Supplementary Fig. 13a, b *bottom*).

Next, to test whether response modality affected the interference effect, we compared the interference effect (incongruent vs. congruent) between the vocal and manual conditions (Fig. 5). In the Stroop task, greater brain activity was observed in the primary auditory cortex in the vocal condition than in the manual condition (Fig. 5a and Supplementary Table 3). On the other hand, relative to the vocal condition, the manual condition showed greater activity in the primary motor cortex and in cerebellar lobules V/VI, which are implicated in right hand movement. As in the Stroop task, the Swimmy task resulted in greater activity in the primary motor cortex and cerebellar lobules V/VI in the manual condition than in the vocal condition (Fig. 5b and Supplementary Table 4). These results suggest that the interference-related activity in the lPFC, PPC, OTC, and cerebellum (crus I/lobule VI and crus II/lobule VIIb) is unaffected by response modalities in both the Stroop and Swimmy tasks.

### Differential cerebro-cerebellar involvement in the Stroop and Swimmy tasks
The observations thus far demonstrated following: (1) the interference effects were independent of response modality in both the Stroop and

Swimmy tasks; (2) the Stroop and reverse-Stroop effects involved similar cross-hemispheric, lateralized, cerebro-cerebellar mechanisms; and (3) the interference effects in the outline and individual tasks in the Swimmy task involved similar bilateral cerebro-cerebellar mechanisms. To directly compare the interference effects between the Stroop task (Fig. 3a) and the Swimmy tasks (Fig. 4a), we contrasted the activation maps between the two tasks. As shown in Fig. 6a and Supplementary Table 5, in the Stroop task, the interference effect was greater in the lPFC and PPC in the left hemisphere and in the crus II in the right cerebellum. On the other hand, the Swimmy task showed a greater interference effect in the PPC in the right hemisphere. These results provide statistical evidence that while the Stroop effect involves left lateral cortical and right cerebellar regions, the Swimmy effect involves bilateral cortical and cerebellar regions.

### The cerebellar involvement in the Stroop effect is associated with language and attentional functions
To functionally characterize the cerebellar involvement, we performed cerebellum-specific imaging analysis (see Methods). An exploratory analysis within the cerebellar atlas space[77] was first performed to identify cerebellar regions involved in the Stroop and Swimmy effects. Then, cerebellar regions were classified into those showing activity (1) only in the Stroop task, (2) only in the Swimmy task, (3) in both tasks, or (4) in neither task. The classified regions were further mapped into ROIs defined by functional parcellation based on task-fMRI data of multi-domain task battery[49].

As shown in Fig. 6b, crus I and II of Region 9[49] showed significant activity only in the Stroop task. Interestingly, this ROI was previously labeled "verbal fluency"[49], a language-related function, and was restricted to the right hemisphere. Notably, this region also showed greater activity in the Stroop task than in the Swimmy task in the whole-brain analysis (Fig. 6a).

In Regions 5 and 6, both of which were labeled "divided attention"[49], bilateral regions in crus I and lobule VI were active in both tasks. In these ROIs, the Stroop task involved the right hemisphere (crus I and lobule VI), whereas the Swimmy task involved both hemispheres (crus I and lobule VI).

These conjunctions and disjunctions within the ROIs reflect well the statistical z-maps for the contrasts of both the incongruent vs. congruent trials (Supplementary Fig. 14) and the Stroop vs. Swimmy tasks (Supplementary Fig. 15). Collectively, the results suggest that cerebellar involvement is associated with language-related functionality in the Stroop task and cognitive functionality in both the Stroop and Swimmy tasks.

Statistical z-maps of the cerebellum were also created for the contrast of the incongruent vs. congruent trials in the vocal condition relative to the manual condition, corresponding to the whole-brain analysis in Fig. 5. Supplementary Fig. 16 shows that Region 2, which was previously labeled "right-hand presses"[49], demonstrated prominent activity in the manual condition, confirming the whole-brain analysis.

### Cross-hemispheric cerebro-cerebellar mechanisms are absent in a large-scale meta-analysis of the Stroop effect
Previous neuroimaging studies of the Stroop effect have reported the involvement of the mPFC/ACC, lPFC, PPC, and OTC[12,16,23,25,27,56–63,67,78]. To evaluate the current observations in reference to previous studies, we compared our results with those of a large-scale meta-analysis of the Stroop effect[62]. As shown in Fig. 7a (top), the meta-analysis revealed broad regions of involvement in the mPFC/ACC, lPFC, PPC, and OTC. In the current study, on the other hand, the involvement was left-lateralized, but overlapped well with that in the meta-analysis. Additionally, our study showed cerebellar involvement that was absent in the meta-analysis. However, the left lateralization in the cortical regions was less pronounced in the Swimmy task (Fig. 7a bottom) than in the

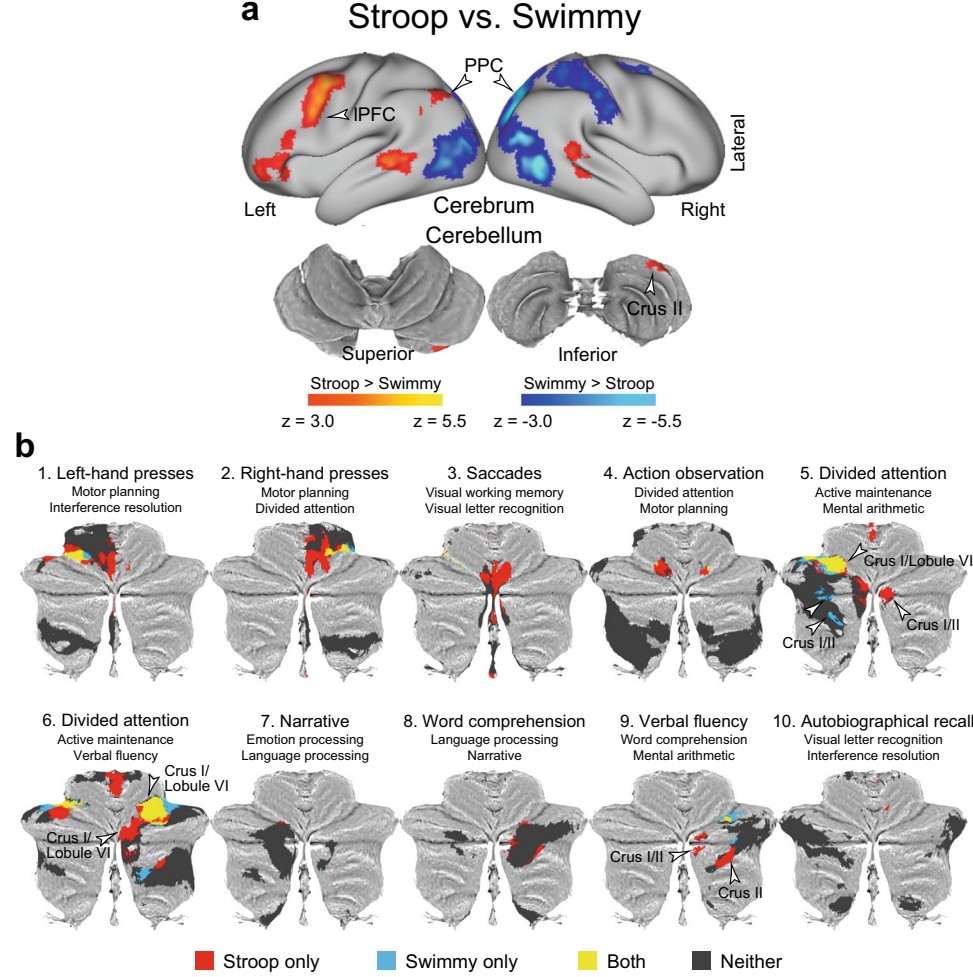

**Fig. 5 | Response and stimulus modality effects. a, b** Response type affects the involvement of sensory-motor regions, but not the fronto-cerebellar loop associated with interference resolution. Statistical maps showing a differential interference effect (incongruent vs. congruent) between the vocal and manual conditions. Stroop task (**a**); Swimmy task (**b**). Hot and cool colors indicate greater interference effect in the vocal and manual conditions, respectively. **c** The Stroop effect predominantly involves the left lPFC and right cerebellum. Statistical maps showing a differential interference effect (incongruent vs. congruent) between the Stroop and Swimmy tasks. Hot and cool colors indicate a greater interference effect in the Stroop and Swimmy tasks, respectively. Formats are similar to those in Fig. 3a.

**Fig. 6 | Direct comparison of the Stroop and Swimmy tasks and functional characterizations of the cerebellum. a** The Stroop effect predominantly involves the left lPFC and right cerebellum. Statistical maps showing a differential interference effect (incongruent vs. congruent) between the Stroop and Swimmy tasks. Hot and cool colors indicate a greater interference effect in the Stroop and Swimmy tasks, respectively. Formats are similar to those in Fig. 3a. **b** The current results for the contrast of the incongruent vs. congruent trials in the Stroop and Swimmy tasks ($P < 0.05$, FWE-corrected across the whole cerebellum based on non-parametric permutation tests) are for the cerebellar ROIs defined by functional parcellation in a previous study mapped onto 2D flat maps of the cerebellum. The functional labels and IDs of the ROIs above the maps were derived in the previous study. Red: Stroop only; blue: Swimmy only; yellow: both Stroop and Swimmy; gray: neither Stroop nor Swimmy.

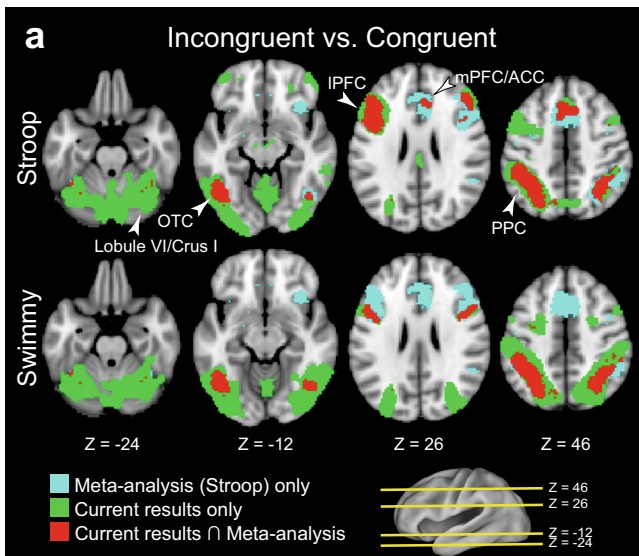

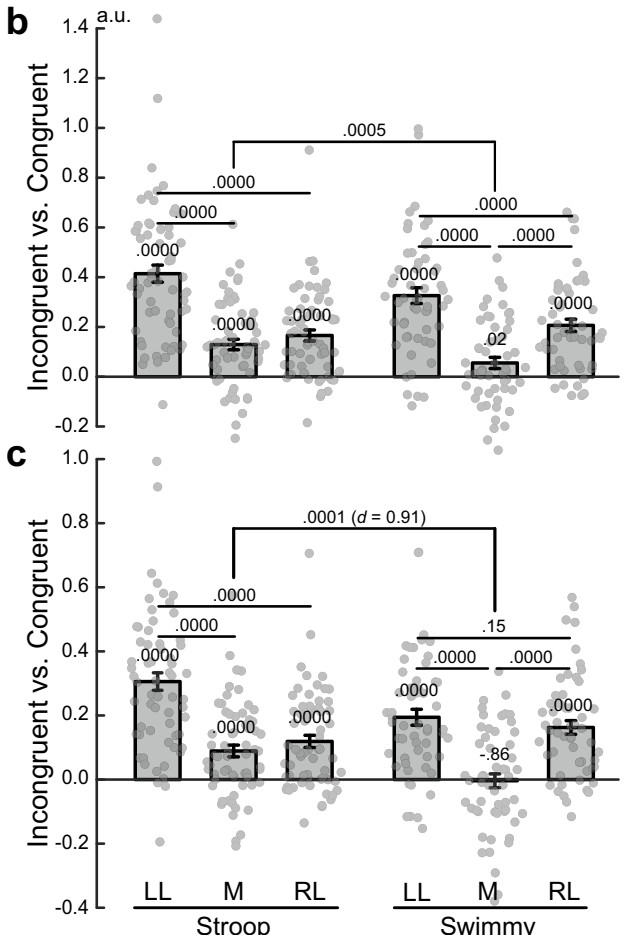

**Fig. 7 | Stroop and Swimmy effects are compared with data from a large-scale meta-analysis. a** Meta-analysis maps of the Stroop effect and the current activation maps of interference effects (incongruent vs. congruent) are overlaid on the transverse sections of structural images. Stroop task (top); Swimmy task (bottom). Cyan: meta-analysis maps only; green: current maps only; red: both maps overlapping. The levels of sections are indicated by Z levels at the bottom and yellow lines on the 3D surface of the brain at the right bottom. Arrowheads indicate anatomical locations. **b** ROI analysis (Stroop: *N* = 63; Swimmy: *N* = 55; independent participants). ROIs were defined as left lateral (LL), medial (M), and right lateral (RL) regions based on *X* axis levels in the meta-analysis maps of the Stroop effect. Stroop task (left); Swimmy task (right). Horizontal and vertical axes indicate ROIs and brain activity, respectively. Rectangular and error bars indicate means and standard errors of the mean across participants, respectively, with individual data overlaid on each rectangular bar. Statistical *p* values are shown on the top (*t* tests, two-tailed, uncorrected). **c** ROI analysis in which ROIs were defined based on the meta-analysis maps of cognitive control. Statistical procedures and formats are similar to those in **b**. The sample size is identical to that in **b**.

[$F_{(1, 116)}$ = 105.6, $P < 0.001$; Fig. 7b], indicating a greater interference effect in the left hemisphere in both the Stroop and Swimmy tasks. The main effect of tasks was not significant [$F_{(1, 116)}$ = 0.42, $P = 0.52$], suggesting that task demands were comparable in the Stroop and Swimmy tasks. Most importantly, the interaction effect of task and laterality was significant [$F_{(1, 116)}$ = 12.9, $P < 0.001$], indicating that left lateralization is greater in the Stroop task than in the Swimmy task. These results were consistently observed in the vocal and manual conditions (Supplementary Fig. 17).

### Hemispheric laterality in cognitive control regions

It is well known that the resolution of cognitive interference requires cognitive control involving the lPFC and PPC[13,16]. We thus performed a similar ROI analysis using a meta-analysis map of cognitive control[62]. Again, the current Stroop effects overlapped well with the meta-analysis map in the left hemisphere, whereas the Swimmy effects overlapped well with the meta-analysis map in both hemispheres (Supplementary Fig. 18a). In the Stroop task, both left and right lateral regions showed significant activity [left: $t_{(62)}$ = 11.1, $P < 0.001$; right: $t_{(62)}$ = 6.3, $P < 0.001$], but the activity was greater in the left hemisphere than in the right [$t_{(62)}$ = 8.4, $P < 0.001$; Fig. 7c]. In the Swimmy task, both hemispheres showed significant activation [left: $t_{(54)}$ = 7.8, $P < 0.001$; right: $t_{(54)}$ = 7.6, $P < 0.001$], and the activity did not differ between hemispheres [left vs. right: $t_{(54)}$ = 1.4, $P = 0.15$]. The hemispheric difference in lPFC activity was greater in the Stroop task than in the Swimmy task [$t_{(116)}$ = 4.9, $P < 0.001$], with a relatively large effect size (Cohen's *d* = 0.91). Post hoc statistical power estimation revealed that the power was high (0.94, alpha rate: 0.001). Because the hemispheric laterality effect in the lPFC was controlled in terms of stimulus verbality (Stroop vs. Swimmy) and laterality (left vs. right), this large effect size and statistical power suggest that greater left lPFC and neocortical activity during the incongruent trials in the Stroop task reflects language functions.

The hemispheric laterality pattern was observed in both the vocal and manual condition. Specifically, the interference effect was greater in the left lateral regions [left vs. right: vocal: $t_{(31)}$ = 4.5, $P < 0.001$; manual: $t_{(30)}$ = 4.9, $P < 0.001$], and in the Swimmy task, the interference effect was observed bilaterally, and the activity did not differ between the left and right hemispheres [left vs. right: vocal: $t_{(26)}$ = 1.2, $P = 0.22$; manual: $t_{(27)}$ = 0.74, $P = 0.47$] (Supplementary Fig. 18b). On the other hand, compared to the lateral regions, the interference effect was weaker in the medial regions [Stroop: left lateral vs. medial: vocal: $t_{(31)}$ = 6.1, $P < 0.001$; manual: $t_{(31)}$ = 5.5, $P < 0.001$; Swimmy: left lateral vs. medial: vocal: $t_{(26)}$ = 5.4, $P < 0.001$; manual: $t_{(27)}$ = 3.4, $P < 0.01$; right lateral vs. medial: vocal: $t_{(26)}$ = 4.2, $P < 0.001$; manual: $t_{(27)}$ = 3.5, $P < 0.01$]. Taken together, these results clearly demonstrate that the Stroop effect is associated with the left lateral cortical regions, while

Stroop task. These comparisons suggest that in our study, the results of the Swimmy task, and not those of the Stroop task, are compatible with the findings of the meta-analysis of the Stroop effect.

To quantitatively evaluate the lateralization, we performed ROI analysis. ROIs were defined as left lateral, medial, and right lateral based on *X* axis coordinates on the meta-analysis map (see Methods). Then, for each ROI, activity magnitudes for the incongruent vs. congruent trials were extracted for the Stroop and Swimmy tasks. A repeated measures ANOVA with task (Stroop and Swimmy) and region laterality (left and right) revealed a significant main effect of laterality

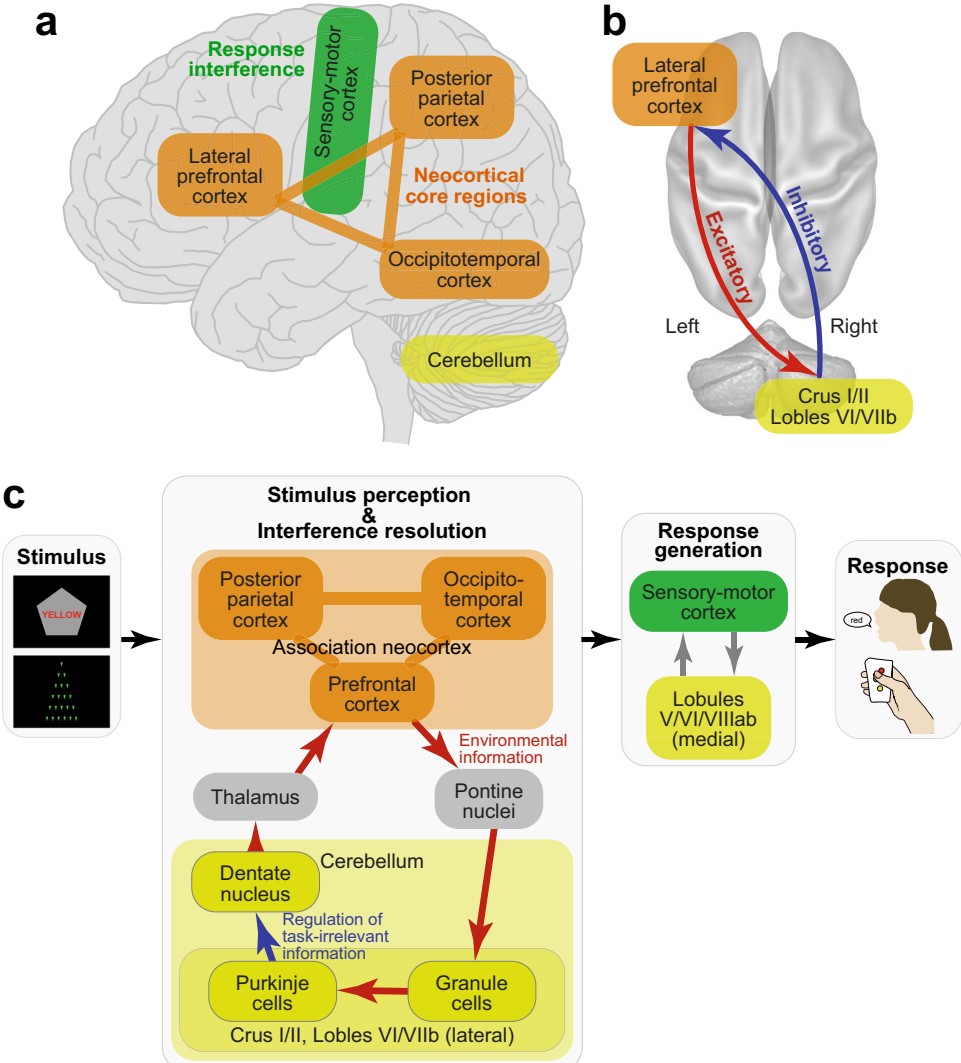

**Fig. 8 | Putative model of the Stroop/Swimmy effects.** Schematic illustrations of cerebro-cerebellar involvement (**a**) and fronto-cerebellar connectivity (**b**) in the Stroop effect as suggested by the current results. **c** Schematic processing diagrams of the Stroop/Swimmy effects from stimulus presentation to response execution. Anatomical structure and functionalities of the front-cerebellar loop are speculated based on previous anatomical and neurophysiological evidence.

bilateral cortical regions are involved in the Swimmy effect, in which stimulus perception does not involve verbal information.

To examine the interference effect in neocortical regions more specifically, the ROIs were divided into lPFC, PPC, OTC, and mPFC/ACC regions (Supplementary Fig. 19). In the Stroop task, the interference effect was greater in the left hemisphere than the right hemisphere throughout the lPFC, PPC, and OTC [lPFC: $t(62) = 6.8$, $P < 0.001$; PPC: $t(62) = 7.9$, $P < 0.001$; OTC: $t(62) = 6.8$, $P < 0.001$]. In the Swimmy task, on the other hand, the interference effect in the left and right hemispheres did not differ for either the lPFC or PPC [lPFC: $t(54) = 0.66$, $P = 0.51$; PPC: $t(54) = 1.34$, $P = 0.19$], whereas in the OTC, the interference effect was greater in the left hemisphere [$t(54) = 4.8$, $P < 0.001$]. The hemispheric difference in lPFC activity was greater in the Stroop task than in the Swimmy task [$t(116) = 4.5$, $P < 0.001$] with a relatively large effect size (Cohen's $d = 0.84$).

These results suggest that left lateralization of cortical involvement in the Stroop effect, but not the Swimmy effect (Fig. 7c, Supplementary Fig. 18b), is derived from activity in the fronto-parietal regions.

This study used multiband imaging that is prone to head motion–induced artifacts, with greater banding with higher multiband factors. To minimize these artifacts, we conducted image processing using ICA-based denoising[79] and motion censoring[80,81], and then re-performed the imaging and connectivity analyses. As shown in Supplementary Figs. 18–20, the results are very similar to those in the original analysis (Figs. 3–5 and 6a), and the original findings were preserved.

## Discussion

The current study examined the neural mechanisms underlying the Stroop effect by manipulating the verbality of the stimulus to be perceived and the response to be made. The resolution of Stroop interference involved the left lateral prefrontal cortex and right cerebellum, but the lateralized involvement was not observed in the non-verbal control task (Fig. 8a). Resolution of the interference by fronto-cerebellar processes involved excitatory signaling from the prefrontal to cerebellar regions and inhibitory signaling from the cerebellar to prefrontal regions (Fig. 8b). These findings were unrelated to the verbality of the response generation. Our results suggest that Stroop interference occurs during the perception of language and color information, and is resolved by the coordinated fronto-cerebellar loop that may regulate goal-relevant information (Fig. 8c).

The neocortical and cerebellar regions are thought to constitute functional-anatomical loops, specifically involving the cerebellar and

pontine nuclei and basal ganglia[41,44,51]. Anatomically, the loop topology has been identified not only between sensorimotor neocortical and ventromedial cerebellar regions, but also between neocortical association regions and dorsolateral cerebellar hemispheres[41,44] (Fig. 8c). A transneuronal tracer study in monkeys identified a loop between the lateral prefrontal cortex and dorsal regions in the lateral cerebellar hemispheres, namely, the crus I/II and lobule VIIb[82]. Interestingly, in our study, homologous regions in the human prefrontal cortex and cerebellum were found to be activated during the incongruent trials (Fig. 3a), suggesting that a fronto-cerebellar loop is involved in the resolution of Stroop interference (Fig. 8).

In relation to task behavior in humans, cortico-cerebellar functional networks have been examined in sensorimotor regions[41,50,51,77,83]. Distinct cortical and cerebellar involvement in voluntary movements is reflected in topographical parcellation of resting-state functional connectivity between the motor cortex and cerebellum[50]. Notably, the lateral prefrontal and dorsolateral cerebellar regions involved in the Stroop interference in the current study also showed strong resting-state connectivity (region 12 in ref. 50).

A classical PET study revealed that language function is associated with the left lateral frontal cortex and right cerebellum, demonstrating a contra-lateral cerebro-cerebellar involvement[41,52]. Neuropsychological studies also showed that damage to the right cerebellum impaired language-related functions[45,46,84,85]. Consistent with these reports, our study demonstrated that a Stroop task involving language functions also relied on cross-hemispheric, lateralized, fronto-cerebellar mechanisms. The functional characterization based on the cerebellum-dedicated imaging analysis (Fig. 6b) suggests that the Stroop effect involves language and attentional functions implemented in the lateral and dorsal parts of the cerebellar cortex.

One interesting finding regarding the directional functional connectivity observed in this study is that the cerebellum sent inhibitory signals to the prefrontal cortex, whereas the prefrontal cortex sent excitatory signals to the cerebellum, a pattern that was enhanced in the incongruent trials (Figs. 3c and 8b). In the cortico-cerebellar loop, Purkinje cells receive excitatory projections from granule cells within the cerebellar cortex, and send inhibitory projections to the deep cerebellar nuclei, which send signals to the thalamus toward neocortical regions[41,44,51] (Fig. 8c). Importantly, the inhibitory projections of the Purkinje cells constitute only one output from the cerebellar cortex. Thus, the inhibitory signals from the cerebellum to the prefrontal cortex observed in our study may reflect the functionality of the Purkinje cells within the cerebro-cerebellar loop (Fig. 8c), although strong fMRI signals in the cerebellum do not reflect Purkinje cell activity alone, and may reflect other neuronal activity of granule cells[41,86]. As such, the strong inhibitory signaling from the cerebellum to the prefrontal cortex may help to filter out task-irrelevant signals derived from visual information that interfered with appropriate task performance in the incongruent trials (Fig. 8c).

Prior neuroimaging studies of the Stroop effect used mainly manual responses and reported involvement of the bilateral and medial prefrontal regions[12,24–26,63,67,78]. In particular, statistically significant activations were reported in right hemisphere regions in some studies[87–89], but other studies identified activations in left hemisphere regions[14,21,22,24,26,28] or in bilateral regions[25,27,29,90,91]. A meta-analysis map of the Stroop effect was created based on relevant studies including those studies[62]. The above findings are inconsistent with the left-lateralized activity in cortical regions in this study, which was found in an exploratory analysis without a priori hypothesis.

Arguments of laterality based only on significant hemisphere-specific activity are not strong because active regions can vary depending on the statistical threshold used. To circumvent this issue, this study directly contrasted activity between the right and left hemispheres, and explored brain regions in which there was greater

activity in one hemisphere than in the other (Supplementary Fig. 4), as in our previous study[68].

Importantly, this discrepancy in the Stroop task was observed not only between the meta-analysis and the current vocal condition, but also between the meta-analysis and the current manual condition. Indeed, our results showed that the left-lateralized involvement of association cortices was independent of response modality (Figs. 5 and 8). In this regard, the current study may highlight a limitation of the meta-analysis approach, specifically that a collection of indecisive results can yield an unclear conclusion, providing a reasonable lesson that meta-analyses should be performed based on decisive studies.

We acknowledge that our contrast of the incongruent vs. congruent trials could involve a facilitation effect (faster response in the congruent trials than in the neutral trials). However, facilitation is not a concomitant of interference[11], the facilitation effect is usually weaker than the interference effect[92], and the interference and facilitation have a common locus[93]. Interestingly, a theoretical study suggested that a low proactive control could lead to reverse facilitation (i.e., faster responses to neutral stimuli than to congruent stimuli)[94]. Thus, we do not think that the facilitation is dominant against the interference in this study.

Previous neuroimaging studies have often used comparisons between the incongruent and congruent trials to minimize oddball effects and to match visual stimuli across conditions[21,24,27,29,88]. Interestingly, two studies compared activity between incongruent trials and neutral trials and between congruent trials and neutral trials, and found that prefrontal activity was higher in the incongruent trials than in both the neutral and congruent trials[16,26], which is consistent with the current study.

A neuropsychological study found that damage to the left lPFC impaired performance in the Stroop task[95], which is consistent with our results. The study also found that damage to the right lPFC impaired performance in a neuropsychological test involving language processing and response inhibition, which agrees with previous studies of response inhibition[96,97].

To characterize and highlight the Stroop effect involving language processing, we developed the Swimmy task as a control task in which the stimulus does not involve language information, like typical flanker tasks[31], but rather behavioral procedures corresponded to those in the current Stroop task. Whereas the Stroop effect showed unilateral neural involvement, the Swimmy effect showed bilateral involvement, which is consistent with prior neuroimaging studies of flanker tasks[32,33,36–38] and the Simon tasks[91,98]. One of these studies directly compared activity during the Stroop task and the Simon task and found greater activity in the left lPFC during the Stroop task[91]. Our study extends this finding by showing that the right cerebellum is also involved specifically in the Stoop task.

Our results also demonstrated that the involvement was independent of response modality. Collectively, these findings indicate that lateralization is dependent on stimulus modality but not on response modality (Fig. 8). Thus, our results suggest that left-lateralized cortical involvement is attributable to the visual word stimulus used in the Stroop task, which is consistent with classical knowledge about language processing in the human brain[17,18]. Additionally, right-lateralized cerebellar involvement in the Stroop effect may reflect perceptual language processing, again consistent with prior neuroimaging and neuropsychological studies[41,45,52].

Samples were independent across the four conditions (vocal Stroop, manual Stroop, vocal Swimmy, and manual Swimmy), and the size of each sample group was comparable to those in standard fMRI studies. All four groups showed robust effects (Figs. 2 and 5 and Supplementary Figs. 2, 5–10, 11, and 14–16). The current results could contain group-specific effects due to the nature of between-group comparisons, but given the sample size and robust results of each

group, the degree of these group-specific effects should be equivalent to those in standard fMRI studies.

Behaviorally, in both the Stroop and Swimmy tasks, the interference effect was greater in the manual conditions than in the vocal conditions (Fig. 2). This pattern is opposite to that observed in previous behavioral studies[15,19,20], possibly due to the high baseline cognitive demand that was further enhanced in the manual conditions because stimulus-response relationships were not straightforward.

Neuroimaging analyses revealed that in the manual conditions, the interference effects were stronger in the primary motor and somatosensory cortices and the motor-related cerebellar regions[41,50,51] in both the Stroop and Swimmy tasks (Figs. 3 and 4). In contrast, in the vocal condition, the primary auditory cortex showed stronger interference effects. These results suggest that the differences in the behavioral interference effects between the vocal and manual conditions are reflected in the greater activity in the motor and somatosensory regions. Interestingly, the association neocortices (lPFC, PPC, OTC) did not show such differential involvement between response modalities, suggesting that these regions contribute to the resolution of cognitive-language interference in both the vocal and manual conditions (Fig. 8). The response modality-dependent interference effects of behavior are attributable to the involvement of the motor and sensory cortices, but not the core mechanisms of interference resolution in the association cortices. Taken together, our imaging results demonstrate that response modality is not a central issue in examining the Stroop effect, as also predicted by classical behavioral studies[11].

Classical behavioral studies have demonstrated an asymmetric interference pattern in reaction times between the color task (Stroop effect) and the word task (reverse Stroop effect); reaction time prolongation is greater in the Stroop effect than in the reverse Stroop effect (see ref. 11 for a review). In the current vocal Stroop condition, we observed an asymmetric pattern in accuracy but not in reaction times, which is likely due to a speed-accuracy tradeoff. Nonetheless, we did not observe such asymmetry in the manual Stoop task, and observed a strong reverse Stroop effect in both the vocal and manual conditions. Because participants perceived the presented colored word and made a judgment about it, the strong reverse interference effect (Fig. 2) may reflect language functions, namely visual word perception and semantic processing.

In the current task, participants first judged the direction of the vertex of a pentagon, and then performed the color task or the word task depending on the vertex direction. This entailed high baseline cognitive demand. Importantly, because the incongruent and congruent trials and the color and word tasks were pseudorandomized in the tasks, the baseline cognitive demand was canceled out when comparing behavior and neuroimaging data between trials and tasks.

Standard Stroop tasks in behavioral and neuroimaging studies have used a blocked design in which participants perform one task (color naming or word reading) continuously. In the block design, because participants are always sure about the upcoming task, they are able to actively maintain task goals (i.e., naming the color by ignoring its word in the color task or reading the word by ignoring its color in the word task), which is referred to as proactive control[99,100]. Importantly, previous studies have demonstrated that enhanced proactive control reduces the interference effect in the Stoop task, both behaviorally[101,102], and neurally[29]. Given this evidence, we used an event-related fMRI design in which trial conditions were intermixed in one task block, and matched the visual stimulus, trial frequency, and baseline cognitive demand across the tasks.

In our task design, participants were unsure before each trial about the task to be performed. As a result, proactive control was unavailable, and only reactive control was possible, with the task goal being reactivated in each trial[99,100]. In this reactive control situation, because processing of word and color is initiated after the presentation of a trial stimulus, the trial imposes high cognitive control.

The longer reaction times in this study were attributable to the reactive control situation. This situation could produce increased processing of the irrelevant dimension and/or reduce the inhibition of the irrelevant information. Thus, the reactive control situation in our task could have yielded a robust reverse Stroop effect in the current study.

Although our behavioral results were not fully consistent with those of classical behavioral studies regarding Stroop effect asymmetry, a greater Stroop effect in the manual condition, and longer reaction times, we note that our neuroimaging results did not reflect these behavioral patterns (Figs. 3 and 5–7). We also note that a previous neuroimaging study[67] reported significant differences in both reaction times and accuracy between the Stroop and reverse Stroop effects, and an asymmetric pattern in accuracy, which is consistent with the current study behaviorally.

The resolution of the Stroop effect refers to the critical operation necessary to achieve a behavioral goal in incongruent trials[99]. The resolution is involved in cognitive processing that is theoretically distinct from interference. Specifically, interference refers to a situation where the behavioral goal is provided and the visual stimulus is perceived, but the critical processing to achieve the goal is not yet complete. This distinction entails that interference (Stroop effect) occurs before its resolution. We acknowledge that the limited temporal resolution of fMRI and our task design did not allow us to differentiate these processes. However, this was not the intent of our study (see also Figs. 1g and 8c).

A key question in this study is how the Stroop effect occurs and is resolved in the brain. This question concerns not only neural correlates (i.e., where in the brain), but also task-related signal processing (i.e., what signals are in the brain). For the former, we performed exploratory analysis across the brain and identified multiple brain regions, including the lPFC and cerebellum. For the latter, we performed directional functional connectivity analysis based on DCM and found inhibitory–excitatory signaling between the lPFC and cerebellum. As such, this study provided neurophysiological evidence to answer the question.

## Methods

### Participants

Written informed consent was obtained from 119 healthy young participants. Each individual participated in one of the four conditions (vocal Stroop: $N = 33$, age range: 18–24, eight females; manual Stroop: $N = 31$, age range 18–23, 11 females; vocal Swimmy: $N = 27$, age range: 18–23, seven females; manual Swimmy: $N = 28$, age range: 18–23, nine females). Experimental procedures were approved by the institutional review board of Keio University and Kochi University of Technology. Participants received 2000 yen for participation. All participants were right-handed, had normal color vision, and spoke Japanese as a native language, and had no history of neurological or psychiatric disorders. The study sample size was determined prior to data collection based on pilot experiments and previous relevant studies. One participant in the vocal Stroop experiments was excluded from analyses due to poor behavioral performance (i.e., low accuracy in the incongruent trials; <0.60).

### Outline of experimental design

A set of participants performed a Stroop task while undergoing functional MRI (Fig. 1c). Another set of participants performed the Swimmy task, an interference task in which the stimulus did not involve verbal information (Fig. 1d). The Swimmy task served as a control condition for the Stroop task. Except for the visual stimulus, identical task procedures were used in the Stroop and Swimmy tasks (Fig. 1e). During the tasks, participants responded verbally or manually (Fig. 1f). Thus, the experiments consisted of two stimulus types (Stroop, Swimmy) and two response types (vocal, manual), entailing a $2 \times 2$ factorial design (Fig. 1g).

We used a between-subject design rather than a within-subject design to minimize the training effect on task performance, since this effect may change the degree of interference in the Stroop and Swimmy tasks[1,11]. In particular, the amount of training for the incongruent trials was equivalent across the four conditions.

## Behavioral procedures

In the Stroop task, a colored word and a gray pentagon were simultaneously presented on the screen in each trial (Fig. 1c). The word, which was composed of Japanese syllabary characters (*kana*), was placed at the center of the pentagon. The colored word and the word color were blue, red, yellow, or green. Participants were required to judge the color or word in each colored word stimulus, depending on which direction in which the pentagon vertex faced. For example, if the vertex pointed upward, participants had to specify the color in the stimulus (color task), and if the vertex pointed downward, they had to specify the word in the stimulus (word task). The color and word in the stimulus were either congruent (color and word were matched) or incongruent (color and word were unmatched). The relationship between the vertex direction and task was counterbalanced across participants.

In contrast to the Stroop task, the stimulus set in the Swimmy task did not involve verbal information. In each trial, a set of small isosceles triangles were presented (Fig. 1d). The orientation of the vertex with the smallest angle was identical for all triangles. The arrangement of the small, individual triangles formed the outline of a larger, homothetic, isosceles triangle. The outline and individual triangles pointed up, right, down, or left. Participants were required to judge the orientation of the vertex of the individual triangles or that of the large triangle outline, depending on the color of the triangles. For example, if the color of the individual triangles was orange, they had to judge the orientation of the large triangle outline, and if the color of the individual triangles was green, they had to judge the orientation of these triangles. The orientation of the outline triangle was either incongruent or congruent with those of the individual triangles. The relationship between the task and the color of the individual triangles was counterbalanced across participants. This task was referred to as the Swimmy task based on an old picture book in which a number of small fish formed the pattern of a large fish[55].

Both the Stroop and Swimmy tasks consisted of two tasks (Stroop: color and word; Swimmy: outline and individual) and two congruency levels (incongruent and congruent), entailing a 2 × 2 factorial design. The four conditions were presented pseudorandomly for 2 s followed by a 2-s fixation period (Fig. 1e). The pseudorandomized design aimed to (1) reduce the proactive control strategy[99] that was proven to reduce behavioral and neural effects in the Stroop task[29,101,102], (2) minimize the oddball effect involving the lateral prefrontal cortex[29], and (3) match visual stimuli and baseline cognitive demands across the tasks.

Participants were instructed to respond as accurately and quickly as possible. Each run consisted of 48 trials, with 12 trials for each of the four conditions. All visual stimuli were presented by E-Prime (Psychology Software Tools, Sharpsburg PA, USA; ver. 2.0.10.356).

Except for the visual stimulus, presentation procedures were identical between the Stroop and Swimmy tasks (Fig. 1e). Participants responded either vocally or manually (Fig. 1f). In the vocal conditions, responses were recorded using an MRI-compatible microphone (FORMRI-III, Optoacoustics, Israel). In the manual conditions, participants responded by pressing a button with their right thumb.

Previous studies of the Stroop effect have often used neutral trials (e.g., a colored circle in the color task and an achromatic word in the word task) rather than congruent trials as a control of the incongruent trials. However, it is well known that perception of visual stimuli involves distinct occipitotemporal regions depending on the modality of visual stimulus[103–105]. Additionally, changes in color or shape were found to drive attentional shifts, which involved the lateral prefrontal cortex[16,106]. Thus, we designed the Stroop and Swimmy tasks such that visual stimuli were identical across conditions (Fig. 1c, d), which enabled strict cross-condition comparisons in neuroimaging analysis.

In both tasks, we also equalized the trial frequency in each condition. This frequency control is important to minimize contamination by the oddball effect, which involves the lateral prefrontal cortex[107]. Indeed, previous neuroimaging studies of the Stroop effect showed that a lower frequency of the incongruent trials enhanced activity in broad brain regions, including the lPFC, PPC, and OTC[29,108], and that this activity might be attributable to the oddball effect. Thus, a neutral trial was not optimal to server as a control condition in order to minimize the oddball effect in terms of color, word, and shape.

## Training procedure

In all four experiments (stimulus: Stroop or Swimmy; response: vocal or manual), participants practiced by performing 96 trials that were presented in two separate runs (48 trials per run). Thus, the practice conditions for the congruent and incongruent trials were matched across the four experiments.

For the manual Stroop task only, because stimulus-response relations were not straightforward (e.g., red–left, blue–right, green–up, yellow–down; Fig. 1 *bottom*), participants received training prior to the task practice in which they were presented with a colored circle patch or a black word, and pressed a corresponding button. This pre-practice training did not involve congruent or incongruent stimuli. In training for the color task, a colored circle patch was presented on a pentagon, and participants pressed a button corresponding to the color. In training for the word task, a color word in black font was presented on a pentagon, and participants were required to press the button corresponding to the color word. Stimulus presentation procedures were identical to those in the Stroop task. The vertex of the pentagon was randomly oriented downward or upward, but did not indicate the task to be performed. Participants performed five runs of the training task. In the other three tasks (i.e., vocal Stroop, manual Swimmy, vocal Swimmy), training for the stimulus-response relationships was unnecessary because the relationships were straightforward. It is important that the training for the stimulus-response relationships in the manual Stroop task did not involve Stroop effects, to avoid contaminating the practice for the congruent and incongruent trials.

## Voice recording and response analysis procedures

Vocal responses were recorded through an MRI-compatible noise-reduction microphone. For each trial, recording was started from the trial onset and stopped at the trial offset. The recording sample rate was 22 kHz. Correct and incorrect responses were sorted manually based on the record data, and trials with completely correct vocal responses within the 2-sec response window were classified as correct trials.

Because automatic MRI noise reduction was insufficient to allow for detection of the onset of vocal responses, an off-line band-path filter was applied to reduce scanner-derived noise (Supplementary Fig. 1). Then, the vocal onset was defined as the first time point the magnitude of the filtered vocal data exceeded a certain threshold, and the reaction time was defined as the latency from the start of the recording to the vocal onset. The bandwidth of the filter and the threshold were determined manually for each participant by adjustment based on visual inspection of the data and detected vocal onsets.

## Behavioral analysis

For each of the Stroop and Swimmy tasks, accuracy and reaction times were calculated for each task (color or word; outline or individual), congruency condition (incongruent or congruent), and response modality (vocal or manual). They were then compared across tasks, congruency conditions, and response modalities within the Stroop or Swimmy task. Statistical tests were performed using SPSS Statistics 25 (IBM Inc. Armonk, NY USA).

Participants completed five runs, each of which consisted of 48 trials (12 trials × 4 conditions). In the vocal Stroop task, participants gave correct responses for $56.5 \pm 3.1$ (mean ± SD) incongruent color trials, $59.8 \pm 0.4$ congruent color trials, $58.0 \pm 2.0$ incongruent word trials, and $59.7 \pm 0.5$ congruent word trials. In the manual Stroop task, they gave correct responses for $53.3 \pm 6.9$ incongruent color trials, $58.9 \pm 1.8$ congruent color trials, $51.7 \pm 6.2$ incongruent word trials, and $58.7 \pm 1.6$ congruent word trials. In the vocal Swimmy task, they gave correct responses for $57.8 \pm 2.7$ incongruent outline trials, $59.6 \pm 0.8$ congruent outline trials, $58.7 \pm 1.5$ incongruent individual trials, and $59.5 \pm 0.7$ congruent individual trials. In the manual Swimmy task, they gave correct responses for $57.4 \pm 2.1$ incongruent outline trials, $59.9 \pm 0.3$ congruent outline trials, $59.0 \pm 1.2$ incongruent individual trials, and $59.7 \pm 0.7$ congruent individual trials.

The numbers of correct trials provided sufficient power for the current imaging analyses, and were comparable not only to the numbers in neuroimaging studies of the Stroop effect[21–23,25–29], but also similar to or greater than those in our previous studies involving uni-variate activation and DCM analyses during cognitive control tasks[68,97,109].

## Imaging procedure
MRI scanning was performed using a 3-T MRI scanner (Siemens Prisma, Germany) with a 64-channel head coil. Functional images were acquired using a multiband acceleration echo-planar imaging sequence [repetition time (TR): 743 msec; echo time (TE): 35.6 msec; flip angle (FA): 48 deg; 72 slices; slice thickness: 2 mm; in-plane resolution: 2 × 2 mm; multiband factor: 8]. One functional run lasted 192 s with 277 volume acquisitions. The initial 10 volumes were discarded for analysis to take into account the equilibrium of long-itudinal magnetization. High-resolution anatomical images were acquired using an MP-RAGE T1-weighted sequence [TR: 1900 msec; TE = 2.52 msec; FA: 9 deg; 176 slices; slice thickness: 1 mm; in-plane resolution: $1 \times 1$ mm$^2$].

## Image preprocessing
MRI data were analyzed using SPM12 software (http://fil.ion.ac.uk/spm/; ver. 6685) running on Matlab 2017a (Mathworks, Inc. Natick, MA USA). All functional images were first temporally aligned across volumes and runs, and then the anatomical image was coregistered to a mean image of the functional images. The functional images were spatially normalized to a standard MNI template with normalization parameters estimated from the anatomical image. The images were then resampled into 2-mm isotropic voxels, and spatially smoothed with a 6-mm full-width at half-maximum (FWHM) Gaussian kernel.

In order to minimize motion-derived artifacts due to vocal responses, functional images were further preprocessed by general linear model (GLM) estimations with motion parameters and MRI signal time courses (cerebrospinal fluid, white matter, and whole-brain), and their derivatives and quadratics as nuisance regressors[110–112] in *fsl_regfilt* in implemented in the FSL suite (http://fmrib.ox.ac.uk/fsl/; ver. 5.0.9). Then residual of the nuisance GLM was used for standard GLM estimations to extract events-related brain activity described below. The same noise reduction procedure was applied to imaging data from all the experiments, including the manual conditions.

To further reduce motion-derived artifacts, we applied ICA-based denoising, implemented by ICA-AROMA[79], and motion censoring[80,81]. For motion censoring, we first calculated motion magnitudes as framewise displacement (FD) values[80]. FD values were $0.192 \pm 0.085$ (mean ± SD) in vocal Stroop, $0.011 \pm 0.021$ in manual Stroop, $0.196 \pm 0.070$ in vocal Swimmy, and $0.103 \pm 0.032$ in manual Swimmy. When FD was thresholded by $0.9$[81,113], the rates of images that exceeded the threshold were $1.2 \pm 3.6\%$ in vocal Stroop, $0.06 \pm 0.1\%$ in manual Stroop, $1.0 \pm 2.4\%$ in vocal Swimmy, and $0.1 \pm 0.4\%$ in manual Stroop. The numbers of participants with images exceeding the threshold by more than 5% were two in the vocal Stroop task, zero in the manual Stroop task, one in the vocal Swimmy task, and zero in the manual Swimmy task. These results clearly indicate greater FD in vocal conditions. However, even in the vocal conditions, the absolute magnitudes were small compared to those in prior studies[80,81]. We next applied motion censoring (scrubbing) based on GLM analysis[113,114]. We created a volume-wise regressor encoding 1 for volumes exceeding FD threshold (0.9) and 0 for others. Then, this effect was regressed out using *fsl_regfilt*.

In this study, because field maps and blip-up/down data were unavailable, we did not perform susceptibility distortion correction. Thus, we note that the accuracy of the spatial identification of the functional locus could be limited, even though we used a modern, standard fMRI scanner with low distortion.

## First-level GLM
Because each participant performed one of the four experimental conditions (vocal Stroop, manual Stroop, vocal Swimmy, or manual Swimmy), the four conditions were separately subjected to first-level GLM analyses. The effects of interest were task dimension (Stroop: color and word; Swimmy: outline and individual) and congruency level (incongruent and congruent). The four trial events (2 tasks and 2 congruency levels) with correct responses were coded separately in a GLM. Error trials were also separately coded in the GLM as a nuisance effect. These trial events were time-locked to the onset of visual stimuli and then convolved with the canonical hemodynamic response function (HRF) implemented in SPM. Then, parameters were estimated for each voxel across the whole brain.

## Group-level statistics
Maps of parameter estimates were first contrasted within individual participants. The contrast maps were collected from all participants, and were subjected to group-level one- or two-sample $t$ tests based on permutation methods (5000 permutations) implemented in rando-mize in the FSL suite. Then voxel clusters were identified using a voxel-wise uncorrected threshold of $P < 0.001$, and the identified voxel clusters were tested for significance with a threshold of $P < 0.05$ corrected by the family-wise error (FWE) rate across the whole brain. This group analysis procedure was validated to appropriately control the false positive rates in a prior study[115]. The peaks of significant clusters were identified and listed in tables. If multiple peaks were identified within 12 mm, the most statistically significant peak was kept. Statistical maps were created using Connectome Workbench (https://www.humanconnectome.org/software/connectome-workbench; ver. 1.4.2) and SUIT[77] (http://www.diedrichsenlab.org/imaging/suit.htm; ver. 3.4). Note that the cerebellar surface maps of the whole-brain analysis are created in the MNI space.

## Activity lateralization analysis
To examine the hemispheric laterality of task-related activity, we explored brain regions showing greater activity than the contra-lateral homologous regions. For each participant, contrast maps (incongruent vs. congruent) were flipped along the $X$ (left-right) axis and were subtracted from the original non-flipped maps on a voxel-by-voxel basis[68]. Then, the group-level statistical significance was tested within the left hemisphere. Note that statistical correction was performed within one hemisphere because the subtracted maps showed sign-flipped symmetry along the $X$ axis (Supplementary Fig. 4).

Exploration was restricted to brain regions in both hemispheres that showed the interference effect ($P < 0.05$ uncorrected). This masking procedure ensured that positive and negative laterality effects indicated greater interference effects in the left and right hemispheres, respectively. We applied this procedure separately to the Stroop and Swimmy tasks, with vocal and manual conditions collapsed.

## Effective connectivity analysis

DCM analysis[72] implemented in SPM12 was performed in order to examine directional functional connectivity associated with the Stroop and Swimmy effects. DCM allows us to explore the effective connectivity among brain regions under the premise that the brain is a deterministic dynamic system that is subject to environmental inputs and that produces outputs based on the space-state model. The model constructs a nonlinear system involving intrinsic connectivity, task-induced connectivity, and extrinsic inputs. Specifically, a model of neural activity was formulated as a linear time-invariant space-state dynamic system,

$$\frac{dx(t)}{dt} = Ax(t) + u(t)Bx(t) + u(t)C \tag{1}$$
$$= [A + u(t)B]x(t) + u(t)C,$$

where $x(t)$ denotes the states of neural activity in $k$ brain regions ($k \times 1$ vector) at time $t$, $u(t)$ denotes inputs to the system from task events at time $t$ (scalar value), $A$ denotes intrinsic connectivity ($k \times k$ matrix), $B$ denotes effective connectivity ($k \times k$ matrix), and $C$ denotes the direct influence of the task variable on neural activity (direct extrinsic input; $k \times 1$ vector). Because the time derivative of neural activity (left side in Eq. 1) is modulated by $[A + u(t)B]x(t)$, the directionality of connectivity is reflected in the $A$ and $B$ matrices. More specifically, the rows and columns of the $A$ and $B$ matrices indicate the target and source of the directionality. The $A$, $B$, and $C$ matrices involved $k^2$, $k(k-1)$, and $k$ parameters, respectively (only non-diagonal elements are parameters for matrix $B$). Thus, the model involved $2k^2$ parameters in total. The unit of the connectivity is arbitrary. Next, the neural activity $x(t)$ in the model (Eq. 1) was transformed as

$$y(t) = \lambda(x(t)) \tag{2}$$

where $\lambda$ denotes a nonlinear function providing fMRI signals from neural activity.

Then, parameters of the nonlinear system ($A$, $B$, and $C$) are estimated based on fMRI time series and task variables/events. The use of a high temporal resolution sequence for functional imaging (TR = 0.743 s) enabled us to collect a large number of scan frames to increase the signal-to-noise ratio of the DCM analysis[68,70,73].

For the Stroop task, we defined ROIs in the left lPFC and the right cerebellum (CER; lobules V/VI and crus II/lobule VIIb) that showed (1) strong activity in the incongruent condition than in the congruent condition (Fig. 3a), and (2) cross-hemispheric lateralization (Fig. 3b). Specifically, the lPFC ROIs were defined within Brodmann area (BA) 44. BA 44 is known to implements multiple cognitive control functions involved in the multiple demand system[116] and to serve as a core prefrontal region for language functions[117].

To ensure that the ROIs were defined independently of the tested data, and to avoid circular analysis[118], the DCM analysis of the vocal Stroop task used the ROIs defined by the manual Stroop task, and vice versa. The exact coordinates in the left lPFC were (−40, 14, 28; $z = 4.67$) in the vocal condition, and (−44, 18, 28; $z = 6.06$) in the manual condition. For the right CER ROIs, the exact coordinates were (crus I/lobule VI: 32, −62, −28; $z = 4.96$) and (crus II/lobule VIIb: 24, −72, −42; $z = 4.85$) in the vocal condition and (crus I/lobule VI: 30, −66, −30; $z = 5.25$) and (crus II/lobule VIIb: 26, −74, −46; $z = 5.72$) in the manual condition.

For the Swimmy task, we defined lPFC and CER ROIs bilaterally, given their bilateral involvement (Fig. 4a, b). As in the Stroop task, ROIs for the vocal condition were defined based on the manual condition, and vice versa to avoid circular analysis[75]. The exact coordinates were (left: −54, 12, 30; $z = 4.00$) and (right: 52, 12, 28; $z = 2.47$) in the vocal condition, and (left: −48, 8, 26; $z = 4.21$) and (right: 46, 8, 30; $z = 3.73$) in the manual condition. For the CER ROIs, the exact coordinates were

(right crus I/lobule VI: 20, −68, −20; $z = 5.31$), (left crus I/lobule VI: − 24, −60, −26; $z = 4.62$), (right crus II/lobule VIIb: 30, −76, −48; $z = 4.37$), and (left crus II/lobule VIIb: −28, −70, −48; $z = 3.53$) in the vocal condition; and (right crus I/lobule VI: 16, −70, −24; $z = 5.44$), (left crus I/lobule VI: − 22, −62, −26; $z = 4.78$), (right crus II/lobule VIIb: 28, −74, −46; $z = 3.95$), and (left crus II/lobule VIIb: −28, −70, −46; $z = 3.80$) in the manual condition.

It should be noted that the all the lPFC ROIs are included in the meta-analysis map of cognitive control in Neurosynth (https://neurosynth.org/analyses/terms/cognitive%20control/; $P < 0.01$, FDR-corrected, uniformity test, minimum $z$ value: 8.4).

To examine the robustness of the connectivity results against ROI definition in the lPFC, lPFC ROIs were re-defined based on the meta-analysis map of cognitive control in Neurosynth (association test). Specifically, the center coordinates of the ROIs were defined as those showing peak $z$-values within BA 44 in the lPFC. The exact coordinates were (−46, 18, 30; $z = 5.6$) and (46, 18, 32; $z = 7.6$) in the left and right lPFC, respectively.

ROI images in lPFC and CER were created as spheres with 6-mm radii centered at these coordinates.

We then tested whether the lPFC region sent (or received) a task-related signal toward (or from) the CER region of the target during the Stroop and Swimmy tasks. Signal time courses (1335 scanning frames) of the lPFC and CER ROIs and regressors in events of interest were extracted from first-level GLMs. The events of interest were the onsets of incongruent and congruent trials. We first separately examined task-related connectivity during the incongruent and congruent trials (relative to baseline). Then we tested whether the connectivity was modulated by task parameters. Specifically, to test whether the task-related connectivity differed between the incongruent and congruent trials, the two trials were coded as a parametrical contrast effect (incongruent: 1; congruent: −1). Likewise, similar parametric effects were examined to test manipulations of tasks [i.e., color and word tasks in the Stroop task (color: 1; word: −1); outline and individual tasks in the Swimmy task (outline: 1; individual: −1)]. The input matrix was U mean-centered.

For each trial effect, causal models were defined as those that differed in modulatory effects (i.e., matrix $B$) among ROIs. We were interested in strengths of effective connectivity between pairs of ROIs (e.g., lPFC and CER), rather than exploration of the model that best fits to the data. Thus, we considered all theoretically possible models. In theory, there are $2^{k(k-1)}$ models that consist of $k$ brain regions. Connectivity matrices reflecting (1) first-order connectivity (i.e., matrix $A$), (2) effective changes in coupling induced by the inputs (i.e., matrix $B$), and (3) extrinsic inputs (i.e., matrix $C$) on MRI signals in ROIs were estimated for each of the models based on DCM analysis implemented in SPM12. When task-related contrast was examined, a parametric regressor (incongruent vs. congruent, color vs. word, or outline vs. individual) was used as an extrinsic effect for effective connectivity between ROIs and ROIs inputs.

In order to estimate the strength of effective connectivity, a Bayesian model reduction method[69] was used. This reduction method reduces the number of models based on model evidence (free energy) and the posterior densities of the models. Specifically, only the models with a minimal posterior density greater than zero were included. These models were then inverted to a fully connected model, and supplemented by second-level parametric empirical Bayes (PEB)[69] to apply empirical priors that remove subjects' variability from each model.

Next, the parameters of these models were estimated based on Bayesian model averaging (BMA)[72] to estimate group-level statistics. Because the current analysis aimed to identify effective connectivity observed as an average across participants, we used a fixed-effect (FFX) estimation assuming that every participant uses the same model. This is in contrast to using a random effect (RFX) estimation assuming

different participants use different models, which is often used to test group differences in effective connectivity[71]. The significance of connectivity was then tested by thresholding at a posterior probability at the 95% confidence interval. We used the uncorrected threshold, because the current analysis aimed to test if connectivity between two specific brain regions was enhanced depending on task manipulation and brain activity, not to identify the model of connectivity among multiple brain regions that best fits to the imaging and behavioral data[68,97,109].

### ROI analysis

In order to examine consistency between the current study and prior neuroimaging studies, ROI analyses were performed. Statistical maps for a meta-analysis of the Stroop effect and cognitive control were obtained from Neurosynth (https://neurosynth.org/analyses/terms/stroop/; https://neurosynth.org/analyses/terms/cognitive control/). The uniformity-test maps were thresholded using a statistical threshold of $P < 0.01$ (FWE-corrected). Then ROIs were defined as left lateral, medial, and right lateral regions in the thresholded maps based on MNI $X$ axis coordinates. Then, for each ROI, activity magnitudes were extracted for the contrast between incongruent and congruent trials from individual participants. The extracted activity magnitudes were then averaged across participants, and compared among ROIs, response modalities, and tasks.

A supplementary ROI analysis was performed to compare activity magnitudes in the lPFC, PPC, OTC, and mPFC/ACC regions (Supplementary Fig. 14). The left lateral, medial, and right lateral ROIs of the meta-analysis maps were further classified into lPFC, PPC, OTC, and mPFC/ACC ROIs based on anatomical locations in the meta-analysis map of cognitive control.

### Imaging analysis of the cerebellum

To examine cerebellar involvement in details, we performed imaging analysis dedicated to the cerebellum based on the SUIT space[77].

Functional images were first realigned and then coregistered to each individual's high-resolution structural image. The individual's cerebellum and brain stem were then extracted from the image based on tissue segmentation and tissue probability calculation implemented in SUIT and SPM12. As previously recommended (https://www.diedrichsenlab.org/imaging/suit_function.htm), for each participant an author (KJ) performed visual inspection and confirmed that the cerebellum and brain stem were appropriately extracted.

Next, the extracted structural image was registered to the SUIT template using nonlinear deformation based on Dartel implemented in SUIT/SPM12. Then, based on the estimated transformation matrix and deformation field as well as the cerebellar mask, functional images were registered to the SUIT space, and resampled into 1-mm isotropic voxels. The registered functional images were spatially smoothed using 6-mm FWHM Gaussian kernel, and motion-related denoising was performed, as in the whole-brain preprocessing based on the MNI space.

Single-level GLM analysis of the cerebellar images was then performed, as in the whole-brain analysis. Group-level analysis was performed similarly, but the FWE rate was corrected within the cerebellum. Then, the cerebellar regions showing significant activity in the contrasts of the incongruent vs. congruent trials in the Stroop and Swimmy task were identified ($P < 0.05$, FWE-corrected). For the regions showing significant activity in the Stroop or Swimmy tasks, we performed a conjunction and disjunction analysis in which the cerebellar regions were classified into those showing activity (1) only in the Stroop task, (2) only in the Swimmy task, or (3) in both tasks, and created conjunction and disjunction images.

To characterize the functionality of the cerebellar regions showing prominent activity, we performed ROI analysis. ROIs were defined by a functional parcellation of the cerebellum based on a multi-domain task battery[49]. Then, the conjunction and disjunction images were

masked by each ROI, and then unfolded to 2D flat maps of the cerebellum.

### Reporting summary

Further information on research design is available in the Nature Portfolio Reporting Summary linked to this article.

## Data availability

The neuroimaging time series and relevant event-onset data generated in this study have been deposited in the Dryad database (https://doi.org/10.5061/dryad.msbcc2g2p)[119]. The behavioral and ROI data generated in this study are provided in the Source Data file. The database of Neurosynth[62] is publicly available (https://neurosynth.org/). Source data are provided with this paper.

## Code availability

A Matlab code to perform estimations of single-level GLMs is provided in the Dryad database via the Zenodo repository (https://doi.org/10.5281/zenodo.7329617)[119].

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

## Acknowledgements

This study was supported by Kakenhi (JPSP: 21H05060, 20K07727, 19H04914, 17K01989, 17H05957, and 17H00891 to KJ; 20H00521, 21K18267 to MT), and a grant from Uehara Memorial Foundation, a grant from Takeda Science Foundation, (PI: Koji Jimura). We thank Ms. Maho Hosono for technical assistance. We also thank Ms. Maoko Yamanaka for administrative assistance.

## Author contributions

M.O., D.T., and K.J. designed the experiment and study. M.O., D.T., R.S., and K.J. collected the data. M.O., T.I., K.T., and K.J. analyzed the data. M.O., M.T., K.N., and K.J. wrote the manuscript.

## Competing interests
The authors declare no competing interests.
