## [Peer Review File · Nature Communications]

The Stroop effect involves an excitatory–inhibitory fronto–cerebellar loopREVIEWER COMMENTS

Reviewer #1 (Remarks to the Author):

This study aimed to examine the neural mechanisms underlying the Stroop effect by manipulating stimulus (verbal, non-verbal) and response (vocal, manual) modality. Authors suggest that the Stroop effect would be resolved by an excitatory-inhibitory loop involving the left prefrontal cortex and the right cerebellum, with an excitatory connectivity from the prefrontal cortex to the cerebellum and an inhibitory connectivity from the cerebellum to the prefrontal cortex.

This is an interesting study on a thorny issue. Four fMRI experiments, each with a satisfactory sample size, are reported. I think this work has some merits, but although neurophysiological data look convincing overall, my enthusiasm for the paper was attenuated by diverse imprecisions, methodological issues, and unexpected results which should be addressed or explained.

1. Absence of Stroop effect asymmetry

No asymmetry was observed (with reaction times) between the Stroop effect, measured in the color naming task, and the reverse Stroop effect (RSE), measured in the word reading task, whereas this outcome is a hallmark of the Stroop paradigm (see MacLeod, 1991, for a review). The RSE is usually not observed in the word reading task. Some studies reported a RSE but only when the conditions of reading were strongly degraded (e.g., Dunbar & MacLeod, 1984; Melara & Mounts, 1993). In this study, a RSE is present under usual reading conditions (i.e., the readability of words was not degraded), for both vocal and manual response modalities. I wondered if a possible explanation could be related to the language. Apparently, there are two writing systems in Japanese: kana (syllabic symbols), which is processed like other phonetic languages such as English, and kanji (a logographic writing system), which is processed like other logographic languages such as Chinese. A reverse Stroop effect was obtained with Japanese kanji but not with Japanese kana by Morikawa (1981; see also Moriguchi & Morikawa, 1998). However, authors used “Japanese syllabary characters”, so results of the word reading task are inconsistent with those commonly reported in the literature. Authors should discuss this unexpected outcome.

It would also be useful to add more details about the specific writing system used in this study.

About Stroop results, authors mentioned (lines 190-191): “our behavioral analysis showed a significant interaction effect of congruency and task (color and word), which is consistent with prior behavioral studies.” This is not exact. The interaction is significant only for accuracy in the verbal task. No interaction between congruency (congruent, incongruent) and task (color naming, word reading) was observed for reaction times (for both vocal and manual responses), which constitute the primary measure in the Stroop paradigm. Furthermore, the authors cite three Stroop studies at the end of this

sentence (Kinoshita, De Wit, & Norris, 2017; Logan, Zbrodoff, & Williamson, 1984; Sharma & McKenna, 1998), but no word reading task was used in these studies.

2. Greater Stroop effect in the manual response modality

The Stroop effect was greater in the manual response modality than in the vocal response modality, for both accuracy and reaction times. Again, this outcome is inconsistent with previous studies. The Stroop effect is known to be greater with verbal responses than with manual responses (e.g., Augustinova, Parris, & Ferrand, 2019; Nielsen, 1974; Redding & Gerjets, 1977; White, 1969).

Lines 126-128: “The interaction effect of response modality (vocal and manual) and congruency was significant [$F(1, 61) = 15.6, P < .001$], indicating that the interference effect was greater in the manual condition, consistent with previous behavioral studies.” This seems incorrect and should be modified. Authors cite three references here to support their results (Kinoshita, De Wit, & Norris, 2017; Logan, Zbrodoff, & Williamson, 1984; Sharma & McKenna, 1998), but these three studies showed a reverse pattern of results, that is, the Stroop effect was lesser with manual responses than with verbal responses. The same mistake is reported in Discussion (lines 514-515).

3. Slow reaction times in the Stroop task

Reaction times in the Stroop task are slow (on average, between 1s and 1.3s), about two times slower than in the classic Stroop task. This could be explained by the specificity of the experimental situation (e.g., determine the direction in which the pentagon vertex faced first, or the fact that word reading and color naming tasks were mixed in the same experiment), but it looks problematic insofar as the Stroop effect depends, at least partially, on the response latency (see e.g., Grégoire, Perruchet, & Poulin-Charronnat, 2014).

The number of trials is also relatively low in the four experiments, and a training could have been done for the vocal Stroop, manual Swimmy and vocal Swimmy, at least to familiarize participants with the task.

Altogether, these unusual results (absence of Stroop effect asymmetry, greater Stroop effect with manual responses, slow reaction times) question the validity of neurophysiological data. If fMRI data reflect atypical behavioral results, the conclusions of this study might be limited.

4. Absence of neutral condition

In this study, the Stroop interference is measured by the difference between the incongruent and the congruent condition, whereas the interference is often measured by the difference between the

incongruent and a neutral/control condition (see e.g., Roelofs, 2010). The difference between the congruent and a neutral/control condition is called facilitation (with usually, faster response times in the congruent condition). Thus, the Stroop effect measured in this study includes interference and facilitation. A neutral condition (e.g., non-words or neutral words) would probably have been more appropriate than a congruent condition to evaluate the Stroop interference. I am not sure this problem can be solved with the current data.

Minor points:

- In the introduction, authors states: "Behavioral studies have also argued that core processing of the Stroop effect involves the generation of the vocal response" (lines 36-37). I am not sure to follow this point. Numerous studies reported a Stroop effect with manual responses, which looks inconsistent with this claim. Please clarify.
- Lines 168 to 185: These analyses include color naming and word reading together (so N = 63)?
- Lines 234-237: this sentence does not look very clear to me. Please add details or clarify.
- Power analysis. Although the sample size in this study looks good to me, it could be useful to perform a power analysis based on previous studies or the number of trials.

Laurent Grégoire

Reviewer #2 (Remarks to the Author):

Okayasu et al. implemented a comparison between a typical color-word Stroop task and a non-verbal (Swimmy) interference task. The authors suggested that the results revealed language specific neural underpinnings in the Stroop task. The tasks are well implemented, and the experiments seem well-powered ($n \sim 30$ in each sub-experiment). However, some critical issues have also arisen during the course of my review. Below I provide several major comments and some minor comments.

1. First of all, although prefrontal regions are consistently found in the Stroop task (incongruent vs. congruent conditions). Whether the activation was lateralized seems to be inconsistent (just to name a few: Bilateral: Leung, et al., 2002; Spielberg, et al., 2011; Peterson, et al. 2002. Right-lateralized: Bench, et al., 1993). This is critical to the current study: Why were lateralized results found in some studies but not the others? Although the current study has attempted to use the "language component" to explain the lateralization, I am not sure if the attempt is successful. First, although the Stroop effect is based on verbal interference, how much the activation in the left prefrontal cortex can be attributed specifically to linguistic information remains elusive. For example, in a patient study, although Cipolotti et al. (2016)

found that performance in a Stroop task relies heavily on the left lateralized frontal regions. Another verbal task, however, relies on the right frontal regions.

2. The key research question and findings are not new. Although I like how the authors target the language component in Stroop, previous MRI studies have compared Stroop with other interferences, which rely less on verbal information. For example, Ven Veen & Carter (2005) have attempted to separate response and semantic interferences in the Stroop task and found that different brain regions were involved in the two interferences. Also, the current study did not significantly deviate from the authors' previous study (Morimoto, et al. 2008) in which they showed that in a Stroop-based flanker task, word interference triggered left inferior frontal gyrus (IFG) activities while color interference triggered right IFG activities. In that study, the authors explained: "...These results suggest that the verbal/nonverbal hemispheric specialization in the IFG can be explained by cognitive control processes per se." I think more work is needed to explain the major advances offered by the current study over previous studies.

3. In multiple places, the authors claimed to find how the Stroop effect is resolved in the brain (and of course where the interference occurs), including in the abstract. I am not sure if I entirely understand this attempt. By resolution, do the authors mean that some neural mechanisms can alleviate the Stroop effect? I think this is certainly not the goal given the experimental design and findings here. So, I guess the authors refer to where the Stroop occurs in the brain. If so, how is that any different from asking where the interference occurs? More explanation on this point is required.

4. Because four separate groups were recruited and performed the Stroop x Swimmy/vocal x manual tasks, I would like to see the authors discuss the possibility of group-specific results and why they chose between-subject over within-subject design in the current study.

Other comments

1. P.11. Lines 259. The authors state "...activity was not as strong in the mPFC/ACC." I am not sure which comparison was made to reach this conclusion.

2. P.13. Lines 306-311. The comparison seems to be between vocal vs. manual. However, the authors reported greater activities in some regions (e.g., primary motor cortex) for both contrasts. That is odd. Please clarify this.

3. P.14. Lines 321-324. The authors conclude that "These results suggest that the roles of the IPFC, PPC, OTC, and cerebellum in the interference effects are independent of response modalities in both the Stroop and Swimmy tasks, and the generation of a response is not critical for resolution of the interference." I think this claim is too strong. Given the results, these regions appeared in both tasks

regardless of the modality, suggesting that modalities did not affect the activities. This is very different from independence.

4. Currently, the number of runs/trials completed for each participant is not described. This will be crucial for future replications.

5. P.27. Lines 671-678. I appreciate that the authors tried to minimize motion-derived artifacts from vocal responses. However, I am not sure I understand how it was achieved. This is critical because, as a reader, I would like to know how head motion caused by vocal responses drove the brain activities.

Reviewer #3 (Remarks to the Author):

In the current manuscript the authors aimed to dissociate the role of language and non-language fronto-cerebellar circuits in resolving Stroop interference. To accomplish this, participants performed either the Stroop task or a novel non-language conflict task (Swimmy task) with vocal or motor responses. The authors found common activation in left lateral prefrontal cortex and right posterior cerebellum for the Stroop task irrespective of the response type. In the Swimmy task, greater left lateral PFC activity was found, but the authors did not find lateralized effects in the cerebellum. In both tasks Dynamic Causal Modeling found that conflict was associated with excitatory activation from PFC to cerebellum and inhibitory activation from cerebellum to PFC. While the topic is quite interesting and the overall approach is novel, I feel that there are major issues that limit my enthusiasm for the paper. Some of these issues can be addressed with major re-analysis, but some issues may not be addressable. I detail the two main issues I identified below. I hope my comments are constructive and aid in improving the impact of the study.

1) The methodological approach lacks rigor and specificity.

a. There are not many details on the registration, but if the authors used SPM normalization without segmentation the cerebellum results may be biased with respect to the cerebellum (Diedrichsen et al., 2009, 2011). From the Group-level statistics section on p. 28, it sounds like SUIT was used to generate figures, but the cerebellar analyses were not carried out in SUIT space. But even when SUIT is used, studies have shown that cerebellar functional parcellations don't necessarily follow lobular boundaries (Bernard et al. 2012 and Buckner et al., 2011, using resting state fMRI, and more recently King et al. 2019 using task based fmri). I would have more enthusiasm for the paper if the authors used SUIT and a functional parcellation e.g., King to define language and non-language posterior cerebellar ROIs.

b. I was disappointed in the lack of spatial specificity in the paper with regards to the lateral PFC. The lateral PFC is large and diverse and there is no acknowledgement of this in the analyses. I find it hard to

interpret the results when they are generalized to such a large region. ROIs were defined for Stroop based on Swimmy activation and vice versa, but I didn't see a clear rationale for this besides wanting to use independent data. I would have preferred to see a theoretically informed ROI, perhaps using the neurosynth ROIs or using data-based parcellations of the frontal lobe (e.g., Neubert et al., 2013; Sallet et al., 2013; Glasser et al., 2016). Another approach that would have improved specificity is using a voxel-based approach for the group-level rather than cluster-based. See Woo et al., 2009 for an excellent discussion of the issues with cluster-based approaches.

c. I was surprised to see no advanced pre-processing given the advanced imaging acquisition parameters. Echo Planar Imaging is prone to geometric distortion and further, multiband imaging is prone to motion induced 'banding artifacts', with greater banding seen with higher multiband factors. A number of preprocessing approaches have been developed to deal with these issues such as distortion correction, automatic and semi-automatic ICA denoising (e.g., ICA-AROMA and ICA-FIX), and motion censoring. Although multiband factor 8 data was collected, none of these preprocessing methods were performed. Unfortunately, there is no mention of field maps or blip-up/down acquisitions being collected, so I am worried that distortion correction could not be applied. Fieldmap-less distortion correction is available in the fmripred advanced preprocessing pipeline, but there are no systematic evaluations of its success in functional imaging to my knowledge.

2) The between subjects design limits the power to compare the Stroop and Swimmy task results. Indeed, the authors didn't perform any analyses that directly compared the groups, focusing instead on qualitative differences between results. Given the large sample size, I expect that the authors had adequate power to perform a group comparison, rather than use qualitative comparisons which may yield flawed inferences. If statistically significant results are not found in the group comparison, the authors could report effect size, which would be more informative for designing follow up studies than non-significant p-values.

a. The abstract doesn't mention the Swimmy task at all, raising the question of whether this task should be reported in this manuscript. However, I find the task to be quite interesting and I would prefer to see a more direct comparison as mentioned above. Relatedly, there isn't much information on the Swimmy task - it seems like a combination of a Simon task and a Local-Global task so the authors should relate the task to these existing tasks as well as to fMRI studies highlighting their neural processes.

Minor issues

1. The authors state that data and code are available upon reasonable request. Given the wide availability of resources to enable open science (OSF, github, openneuro, etc), I find this data availability statement to be unacceptable. By placing code and data in repositories the authors both help to ensure that data and code are not lost and/or the knowledge of how to interpret the code is not lost. I encourage the authors to openly share their data and code.

2. I noted a couple of typos. Crus is misspelled as crura in 2 places and Flanker is misspelled franker one time that I noted.

Signed,

Joseph Orr, PhD

Texas A&M University

Response to Reviewer Comments

Manuscript by Okayasu et al. (original manuscript number: NCOMMS-22-06757-T)

Title: An excitatory–inhibitory fronto–cerebellar loop resolves the Stroop effect.

We wish to thank the editors and reviewers for handling and evaluating our original manuscript. We were gratified to see the positive reactions to our manuscript. Additionally, the reviewers expressed some concerns and provided helpful and constructive suggestions regarding the original manuscript. In accordance with the reviewers' comments, we thoroughly revised the manuscript; we performed new behavioral and imaging analyses, added new figures, rearranged figures, and clarified the key question, novel findings, results interpretation, and methodological issues of the study. The important revisions are listed below:

- 1) We performed a new analysis to show the robust laterality of the activity and its role in language functions in response to the comments of reviewer #2. The results are shown in a new figure (Fig. 7c) and described in the Results.
- 2) To functionally characterize cerebellar involvement in the Stroop effect, we performed a new analysis in response to the comments of reviewer #3. The results are shown in new figures (Fig. 6b; Supplementary Figs. 14–16) and described in the Methods and Results.
- 3) To reduce head motion–related artifacts, we performed advanced preprocessing of functional images in response to the comments of reviewer #3. The results of activation and connectivity analyses are shown in new figures (Supplementary Figs. 19–21).
- 4) To specify the functionality of the IPFC in response to the comment of reviewer #3, we performed a new connectivity analysis by defining IPFC ROIs based on a meta-analysis. The results are shown in new figures (Supplementary Figs. 5 and 10). We have also clarified the definition of ROIs in the Methods.
- 5) By adding new figures (Figs. 1b and 8b), we now highlight a cerebro–cerebellar loop as a key issue and novel finding of the study in response to the comments of reviewer #2. Additionally, we describe the novelty of our findings in the Results and Discussion sections.
- 6) We describe in greater detail that our behavioral results are reasonably interpreted in line with previous studies of the Stroop effect in the Methods, Results, and Discussion sections, in response to the comments of reviewer #1.
- 7) We describe rationale of our task design, which was optimized for our event-related neuroimaging task, in the Methods and Discussion sections, in response to the comments of reviewer #1.

In this document, we present point-by-point responses to the reviewers' comments and indicate where we have made relevant revisions in the manuscript. Revisions are highlighted in red in the revised manuscript.

We hope that these revisions satisfactorily address the reviewers' concerns. The page numbers refer to those in the revised manuscript unless otherwise mentioned.

Reviewer #1

Reviewer's comment #1:

1. Absence of Stroop effect asymmetry

No asymmetry was observed (with reaction times) between the Stroop effect, measured in the color naming task, and the reverse Stroop effect (RSE), measured in the word reading task, whereas this outcome is a hallmark of the Stroop paradigm (see MacLeod, 1991, for a review). The RSE is usually not observed in the word reading task. Some studies reported a RSE but only when the conditions of reading were strongly degraded (e.g., Dunbar & MacLeod, 1984; Melara & Mounts, 1993). In this study, a RSE is present under usual reading conditions (i.e., the readability of words was not degraded), for both vocal and manual response modalities.

Authors' response:

We appreciate the reviewer's crucial point about the reverse Stroop effect and the asymmetric behavioral pattern between the Stroop and reverse Stroop effects. As the reviewer pointed out, reaction times did not demonstrate the asymmetric pattern, but did show strong reverse Stroop effects. We acknowledge that the strong reverse Stroop effect has not typically been observed in classical standard behavioral studies (MacLeod 1991).

We wish to note that we observed an asymmetric pattern in the accuracy of the vocal Stroop task (lines 127–129). This interaction effect was not statistically significant for reaction times, but this was seemingly due to a speed–accuracy tradeoff. We understand that asymmetry is usually observed in reaction times, but we would like to note that previous studies have reported accuracy effects as indicators that co-occurred with the Stroop effect and reflected the trends in the reaction times (e.g., Glaser and Glaser, 1982; Logan and Zbrodoff, 1984; Kinoshita et al., 2017). Thus, our behavioral results are not incompatible with classical studies of the Stroop tasks (MacLeod, 1991).

Two behavioral studies listed by the reviewer (Dunbar and MacLeod, 1984; Melara and Mounts, 1993) provide important evidence about the reverse Stroop effect. As the reviewer pointed out, these studies demonstrated that when baseline cognitive and perceptual demands were increased, the reverse Stroop effect became stronger. Interestingly, increased cognitive demand for response generation also enhanced the reverse Stroop effect (Durgin 2000, 2003). These collective results suggest that baseline cognitive demand plays an important role in the reverse Stroop effect.

The task situations in these previous studies are highly consistent with those in the current study in terms of the baseline cognitive demand. In the current task, participants first judged the direction of the vertex of a pentagon, and then performed the color task or the word task depending on the vertex direction (Fig. 1c). Thus, the reverse Stroop effect in the current study is attributable to the high baseline cognitive demand due to the judgment of the task to be performed, a situation similar to those in the previous studies (Dunbar and MacLeod 1984; Melara and Mount, 1993; Durgin 2000, 2003). We note that the participants were instructed and trained to always judge the vertex direction first and then perform the color or word task, as appropriate. Importantly, because the incongruent and congruent trials and the color and word tasks were pseudorandomized in our task

(Fig. 1d), the baseline cognitive demand was cancelled out when comparing behavioral data between trials and tasks. The same applies to the neuroimaging data.

A fundamental question then arises about why we used a task design when baseline cognitive demand is high. A critical reason is that we aimed to optimize our task design for a neuroimaging experiment. We describe this reason in detail below.

Standard Stroop tasks in behavioral and neuroimaging studies have used a blocked design in which participants continuously perform a single task (color naming or word reading). In the block design, because participants are always sure about the upcoming task, they are able to actively maintain task goals (i.e., naming the color by ignoring its word in the color task or reading the word by ignoring its color in the word task), which is referred to as proactive control (Jimura et al., 2010; Braver, 2012). Previous studies have demonstrated that enhanced proactive control reduces the interference effect in the Stroop task, both behaviorally (Kalanthoff et al. 2015; Spinelli et al. 2021) and neurally (Leung et al. 2000). Indeed, in our pilot experiment, we used a blocked design as in standard studies, and instructed participants to fixate their eyes on the center cross to avoid saccadic movements that involve the lateral prefrontal cortex. We also matched the frequency of trials across conditions to reduce oddball effects that also involve the lateral prefrontal cortex (see also our response to comment #4).

In this pilot experiment, proactive control was strong, possibly because fixating to the screen center made participants easily able to ignore the periphery of the stimulus, which entailed that they perceived the color without perceiving the word. Moreover, the equivalent frequency of the incongruent and congruent trials also enhanced proactive control, like in the previous neuroimaging studies of the Stroop effect (Leung et al., 2000; Kalanthoff et al., 2015; Spinelli et al., 2021). We found that the neural and behavioral effects of the Stroop interference were weak in the block design. Given these weak effects in our imaging pilot study, we used an event-related fMRI design in which trial conditions were intermixed in one task block; in our task, visual stimulus, trial frequency, and baseline cognitive demand were matched across the task conditions (see also our response to comment #4 below).

In contrast to proactive control, reactive control refers to reactivation of the task goal in each trial (Jimura et al. 2010; Braver et al. 2012). In our task design, participants were unsure before each trial about the task to be performed; therefore, proactive control was unavailable and only reactive control was possible. In a reactive control situation, because processing of word and color is initiated after the presentation of a trial stimulus, the trial imposes high cognitive control. Therefore, this task situation modulated predominance of processing word and color, as demonstrated in previous studies (Dumber and MacLeod, 1984; Melara and Mounts, 1993).

Interestingly, the interpretation of our results fits well with a theoretical account of the Stroop effect formulated and revised by previous studies (Cohen et al., 1990; Gregoire et al., 2014). Specifically, the degree of the processing predominance for color and word becomes equivalent (Situation 3 in Fig 2 of Gregoire et al. 2014), and both of the Stroop and reverse Stroop effects become

strong. We consider that this situation occurred in the current Stroop task, and our results are consistent with those in the Stroop literature.

Taken together, the reactive strategy increased cognitive control demand, which is attributable to the strong reverse Stroop effect in the current study. We hope that our clarification of the task design and interpretation of the strong reverse Stroop effect reasonably address the reviewer's concern. We clarified these points in the Discussion (lines 657–676; 681–703) and Methods (lines 806–810).

Reviewer's comment #1 (cont.):

I wondered if a possible explanation could be related to the language. Apparently, there are two writing systems in Japanese: kana (syllabic symbols), which is processed like other phonetic languages such as English, and kanji (a logographic writing system), which is processed like other logographic languages such as Chinese. A reverse Stroop effect was obtained with Japanese kanji but not with Japanese kana by Morikawa (1981; see also Moriguchi & Morikawa, 1998). However, authors used "Japanese syllabary characters", so results of the word reading task are inconsistent with those commonly reported in the literature. Authors should discuss this unexpected outcome. It would also be useful to add more details about the specific writing system used in this study.

Authors' response:

We appreciate the reviewer's interesting point about the Stroop effect in Japanese *kana* and *kanji*. We used *kana* in our task stimuli. We now have clarified this point in the Methods (line 779). Our results may not be suitable to compare to those in the studies of Morikawa (1981) and Moriguchi and Morikawa (1998), because we did not compare the effects of *kanji* and *kana*. Nonetheless, we once again wish to note that even though we used *kana* in our stimulus, we observed the reverse Stroop effect possibly because the proactive strategy was unavailable in our study. Additionally, when performing the Stroop task, participants perceived the colored word and then made a judgment about it, and therefore the strong reverse interference effect in the current study may be related to language functions. We have clarified this point in the Discussion (lines 676–679).

Reviewer's comment #1 (cont.):

About Stroop results, authors mentioned (lines 190-191): "our behavioral analysis showed a significant interaction effect of congruency and task (color and word), which is consistent with prior behavioral studies." This is not exact. The interaction is significant only for accuracy in the verbal task. No interaction between congruency (congruent, incongruent) and task (color naming, word reading) was observed for reaction times (for both vocal and manual responses), which constitute the primary measure in the Stroop paradigm. Furthermore, the authors cite three Stroop studies at the end of this sentence (Kinoshita, De Wit, & Norris, 2017; Logan, Zbrodoff, & Williamson, 1984; Sharma & McKenna, 1998), but no word reading task was used in these studies.

Authors' response:

In response to the reviewer's comment, we redrafted the sentence to indicate that the interaction effect was observed in the accuracy of the vocal response (lines

201-203). We also deleted the citations for these previous studies.

Reviewer's comment #2:

2. Greater Stroop effect in the manual response modality

The Stroop effect was greater in the manual response modality than in the vocal response modality, for both accuracy and reaction times. Again, this outcome is inconsistent with previous studies. The Stroop effect is known to be greater with verbal responses than with manual responses (e.g., Augustinova, Parris, & Ferrand, 2019; Nielsen, 1974; Redding & Gerjets, 1977; White, 1969).

Authors' response:

As we noted above, an important characteristic of our task is the pseudorandomized trial conditions across congruency and tasks (color and word). On the other hand, the previous studies that the reviewer listed (Augustinova, Parris, & Ferrand, 2019; Nielsen, 1974; Redding & Gerjets, 1977; White, 1969) used a blocked design where proactive control was available.

The pseudorandomized condition design is crucial for two reasons: 1) to reduce the oddball effect involving the lateral prefrontal cortex, and 2) to minimize proactive control that makes it possible to ignore the irrelevant stimulus modality (i.e., word in the color task and color in the word task). Instead, it enhanced baseline cognitive demand (see also our response to comment #1 above). The greater interference effect in the manual condition in the current study may have been due to the high baseline cognitive demand, which was further enhanced in the manual condition. This could have occurred because stimulus–response relationships were not straightforward and needed additional training prior to task practice. As such, although the degree of the interference between vocal and manual conditions was reversed compared to the previous studies, our behavioral results can be reasonably interpreted.

We wish to note that our imaging results demonstrate that response modality is not a central issue in examining the Stroop effect, as also forecasted by classical behavioral studies (see MacLeod, 1991, for a review). Indeed, in our imaging analysis, the congruency effect involved neocortical association regions including the IPFC, PPC, and OTC in the left hemisphere and lateral cerebellar cortices in the right hemisphere (Fig. 3a). This congruency effect was commonly observed in the manual and vocal responses (Supplementary Fig. 2). On the other hand, the response modality effect did not involve the neocortical association regions or lateral cerebellar regions, but did involve primary motor, auditory, and sensory regions in the neocortex and cerebellum (Supplementary Fig. 13). Most importantly, the difference in the interference effect between the vocal and manual responses was associated with the motor and sensory regions (Fig. 6a); this result indicates that the greater behavioral interference in the manual response (Fig. 2) was reflected in the activity in the motor-related neocortical and cerebellar regions but not in the activity in the neocortical association regions critical for the resolution of the Stroop effect (i.e., IPFC, PPC, and OTC; Fig. 8a).

We hope our interpretation and results reasonably address the reviewer's concern. We now describe these points in the Discussion (lines 636–640, 653–655).

Reviewer's comment #2 (cont.):

Lines 126-128: “The interaction effect of response modality (vocal and manual) and congruency was significant [$F(1, 61) = 15.6, P < .001$], indicating that the interference effect was greater in the manual condition, consistent with previous behavioral studies.” This seems incorrect and should be modified. Authors cite three references here to support their results (Kinoshita, De Wit, & Norris, 2017; Logan, Zbrodoff, & Williamson, 1984; Sharma & McKenna, 1998), but these three studies showed a reverse pattern of results, that is, the Stroop effect was lesser with manual responses than with verbal responses. The same mistake is reported in Discussion (lines 514-515).

Authors’ response:

The reviewer is correct that our results are not consistent with those in the previous studies regarding the degree of interference between the vocal and manual responses. Accordingly, we deleted the relevant phrase in the Results (line 132), and, as we noted above, we now discuss this discrepancy in the Discussion (lines 636–640).

Reviewer’s comment #3:

3. Slow reaction times in the Stroop task

Reaction times in the Stroop task are slow (on average, between 1s and 1.3s), about two times slower than in the classic Stroop task. This could be explained by the specificity of the experimental situation (e.g., determine the direction in which the pentagon vertex faced first, or the fact that word reading and color naming tasks were mixed in the same experiment), but it looks problematic insofar as the Stroop effect depends, at least partially, on the response latency (see e.g., Grégoire, Perruchet, & Poulin-Charronnat, 2014).

Authors’ response:

We appreciate the reviewer’s important point regarding our task design and the reaction times. As the reviewer pointed out, the longer reaction times in our study are attributable to the recognition of the pentagon vertex and to judgment about the task to be performed. We now describe this point in the Results (lines 160–162) and Discussion (lines 693–698).

As the study by Grégoire et al. (2014) pointed out, the degree of interference of the Stroop task is usually evaluated by reaction times. Importantly, the degree of interference is represented by the prolongation of reaction times in the incongruent trials relative to the congruent trials. In our study, the reaction-time prolongation was 117 ± 67 (mean \pm SD) ms in the color task and 140 ± 80 ms in the word task in the vocal condition, and 294 ± 145 ms in the color task and 263 ± 123 ms in the word task in the manual condition. These interference magnitudes are comparable to those in the classical and previous studies of the Stroop effect (Table 1 of Sharma and McKenna, 1998; Logan et al., 1984; Kinoshita et al., 2017). We now describe this result in the Results (lines 162–166).

Despite the longer absolute reaction times, we hope that the reviewer’s concern is addressed by the result that the degree of interference in the incongruent trials was comparable to those in classical studies.

Reviewer’s comment #3 (cont.):

The number of trials is also relatively low in the four experiments, and a training

could have been done for the vocal Stroop, manual Swimmy and vocal Swimmy, at least to familiarize participants with the task.

Authors' response:

We acknowledge that the number of trials was relatively low for a behavioral study, but note that it was comparable to those in neuroimaging studies of the Stroop effect (e.g., Taylor et al., 1997; McKeown et al., 1998; Banich et al., 2000; Leung et al., 2000; Barch et al., 2001; Zysset et al., 2001; Egner and Hirsch, 2005; Freund et al., 2021). Moreover, the number of correct trials was comparable to or greater than those in our previous studies involving univariate activation and DCM analyses during cognitive control tasks (Tsumura et al., 2021, 2022a, 2022b), indicating that the current analyses have sufficient power.

Specifically, all participants completed five runs, each consisting of 48 trials (12 trials \times 4 conditions) (i.e., 60 trials per condition in total). In the vocal Stroop task, participants gave correct responses for 56.5 ± 3.1 (mean \pm SD) incongruent color trials, 59.8 ± 0.4 congruent color trials, 58.0 ± 2.0 incongruent word trials, and 59.7 ± 0.5 congruent word trials. In the manual Stroop task, they gave correct responses for 53.3 ± 6.9 incongruent color trials, 58.9 ± 1.8 congruent color trials, 51.7 ± 6.2 incongruent word trials, and 58.7 ± 1.6 congruent word trials. In the vocal Swimmy task, they gave correct responses for 57.8 ± 2.7 incongruent outline trials, 59.6 ± 0.8 congruent outline trials, 58.7 ± 1.5 incongruent individual trials, and 59.5 ± 0.7 congruent individual trials. In the manual Swimmy task, they gave correct responses for 57.4 ± 2.1 incongruent outline trials, 59.9 ± 0.3 congruent outline trials, 59.0 ± 1.2 incongruent individual trials, and 59.7 ± 0.7 congruent individual trials. We now describe this point in the Methods (lines 880–895).

Regarding the training issue, we would like to note that prior to the scan experiments, participants underwent practice trials (96 trials, which corresponded to two functional runs) in all four tasks (vocal Stroop, manual Stroop, vocal Swimmy, and manual Swimmy). Thus, the number of the practice trials was matched across the four tasks (lines 838–841).

For the manual Stroop task only, because stimulus–response relations were not straightforward (e.g., red–left, blue–right, green–up, yellow–down), participants received training prior to the task practice in which they were presented with a colored circle patch or a black word, and pressed the corresponding button. This pre-practice training did not involve congruent or incongruent stimuli. We have now clarified this point in the Methods (lines 842–846, 855–857).

Reviewer's comment #3 (cont.):

Altogether, these unusual results (absence of Stroop effect asymmetry, greater Stroop effect with manual responses, slow reaction times) question the validity of neurophysiological data. If fMRI data reflect atypical behavioral results, the conclusions of this study might be limited.

Authors' response:

We again appreciate the reviewer's critical points about our behavioral results. As we noted in response to comments #1 and #2 above and to comment #4 below, although the behavioral results are not fully compatible with those in classical behavioral studies, our results can be reasonably interpreted given that our task design was optimized for a neuroimaging study.

Importantly, as we indicated in our response to comment #2 above and show in Figs. 3 and 5–7, critical neuroimaging results did not reflect these behavioral patterns. We thus believe that our neuroimaging results are valid. We also note that a previous neuroimaging study of the Stroop and reverse Stroop effects reported significant differences in both reaction times and accuracy in both effects, as well as an asymmetric accuracy pattern (Song and Hakoda 2015), which is consistent with our study. While that study reported brain activity differences between the Stroop and reverse Stroop effects, false positive rates were not appropriately controlled (i.e., p-values were not corrected for multiple comparisons). By contrast, our results included a robust statistical test for this comparison in which the false-positive rates were appropriately controlled using non-parametric permutation testing (Eklund et al., 2016). We now describe this point in the Discussion (lines 704–710). We hope our responses and revisions satisfactorily address the reviewer’s concern.

Reviewer’s comment #4:

4. Absence of neutral condition

In this study, the Stroop interference is measured by the difference between the incongruent and the congruent condition, whereas the interference is often measured by the difference between the incongruent and a neutral/control condition (see e.g., Roelofs, 2010). The difference between the congruent and a neutral/control condition is called facilitation (with usually, faster response times in the congruent condition). Thus, the Stroop effect measured in this study includes interference and facilitation. A neutral condition (e.g., non-words or neutral words) would probably have been more appropriate than a congruent condition to evaluate the Stroop Interference. I am not sure this problem can be solved with the current data.

Authors’ response:

We appreciate the reviewer’s point about the control condition of our task. We acknowledge that behavioral studies often use neutral conditions such as colored pseudowords, unrelated words, and scrambled images in the color naming task, and black and gray words in the word reading tasks (MacLeod 1991). However, as MacLeod (1991) pointed out, facilitation is not a concomitant of interference, and the facilitation effect is weaker than the interference effect (pp. 175; see also Glaser and Dungenhoff, 1984). Consistent with this point and with behavioral evidence, previous neuroimaging studies compared activity between incongruent and neutral trials and between congruent and neutral trials and found that prefrontal activity was higher in the incongruent trials than in both the neutral and congruent trials (Carter et al., 1995; Zysset et al., 2001), which is also compatible with our study.

In cognitive neuroscience, it is well known that the perception of different colors involves distinct cortical regions (e.g., Zeki et al., 1991; Morita et al., 2004). It is also well known that the perception of visual stimuli involves specific occipitotemporal regions depending on stimulus modality (e.g., color and word; Zeki et al., 1991; Dehaene et al., 2011). Moreover, changes in color or shape drive attentional shifts (Carter et al., 1995) involving the IPFC (e.g., Serences and Yantis, 2007). Given this prior understanding of visual perception, the current Stroop task was designed such that visual stimuli were identical across conditions, which

enabled strict cross-condition comparisons in the neuroimaging analysis.

Moreover, we equalized the trial frequency in each condition in both the Stroop and Swimmy tasks. This frequency control is important to minimize contamination by the oddball effect, which involves the LPFC (e.g., Brazdil et al., 2005). In the current study, inclusion of the neutral condition would make it impossible to control the trial frequency in terms of color, word, and shape. Critically, in some neuroimaging studies of the Stroop effect that used a blocked design (e.g., Leung et al., 2000; Freund et al., 2021), the frequencies of the incongruent and congruent trials were not equal. Indeed, a lower frequency of the incongruent trials enhanced activity in broad brain regions, including the LPFC, PPC, and OTC (Leung et al., 2000; Melcher and Gruber, 2006). Thus, the enhanced activity in the incongruent trials in the previous studies might be attributable to the oddball effect.

We acknowledge that our contrast of the incongruent vs. congruent trials involves the facilitation effect (faster response in the congruent trials than in the neutral trials). However, as stated above, because the facilitation is not a concomitant of interference, and the facilitation effect is weaker than the interference effect (MacLeod, 1991; Glaser and Dangelhoff, 1984), we do not think that the facilitation is dominant against the interference in our study.

Notably, because of the oddball effect and stimulus perception mechanisms in the brain, neuroimaging studies have often used comparisons between the incongruent and congruent trials (e.g., Pardo et al., 1990; Leung et al., 2000; Egner and Hirsch, 2005; Schulte et al., 2009; Freund et al., 2021).

Taken together, we hope that the reviewer's concern is addressed by 1) optimization of our task design for neuroimaging, 2) the nature of the facilitation, and 3) the agreement of our neuroimaging results with prior neuroimaging studies. We clarified these points in the Discussion (lines 590–600) and Methods (lines 819–835).

Minor points:

Reviewer's comment #1:

- In the introduction, authors states: "Behavioral studies have also argued that core processing of the Stroop effect involves the generation of the vocal response" (lines 36-37). I am not sure to follow this point. Numerous studies reported a Stroop effect with manual responses, which looks inconsistent with this claim. Please clarify.

Authors' response:

As the reviewer pointed out, many studies used manual responses to examine the Stroop effect, whereas the original study (Stroop, 1935) used vocal responses. Some behavioral studies of the Stroop effect have suggested that the vocal response specifically involves language processing (Kinoshita et al., 2017), that the vocal and manual responses employ distinct task strategies (Logan et al., 1984), and that language components are less involved in the manual response (Sharma and McKenna 1998). We acknowledge that these studies did not argue that core processing of the Stroop effect involves generation of vocal responses. Accordingly, we have revised the sentence (lines 36–38).

Reviewer's comment #2:

- Lines 168 to 185: These analyses include color naming and word reading together (so N = 63)?

Authors' response:

Yes, the color naming and word reading task were collapsed here. We now describe this point in the Results (lines 181–182).

Reviewer's comment #3:

- Lines 234-237: this sentence does not look very clear to me. Please add details or clarify.

Authors' response:

In this functional connectivity analysis based on DCM, we defined ROIs in the IPFC and cerebellum. When defining the ROIs, we aimed to avoid circular analysis by ensuring that definition data and tested data were independent (Kriegeskorte et al., 2009). To this end, for the vocal condition, we defined ROIs based on the activation map of the manual condition (Supplementary Fig. 2b). Conversely, for the manual condition, we defined ROIs based on the activation map of the vocal condition (Supplementary Fig. 2a). Specifically, we identified peak activation coordinates in the IPFC and cerebellum. The exact coordinates in the IPFC were (−40, 14, 28; $z = 4.67$) in the vocal condition, and (−44, 18, 28; $z = 6.06$) in the manual condition. Likewise, the exact coordinates in the cerebellum were (crus I/lobule VI: 32, −62, −28; $z = 4.96$), and (crus II/lobule VIIb: 24, −72, −42; $z = 4.85$) in the vocal condition; (crus I/lobule VI: 30, −66, −30; $z = 5.25$) and (crus II/lobule VIIb: 26, −74, −46; $z = 5.72$) in the manual condition. Then, the ROI images were created as spheres with 6-mm radii centered at the coordinates. A similar procedure was applied to define the ROIs for the DCM analysis of the Swimmy task (Fig. 4c)

We have now clarified this point in the Results (lines 246–247) and Methods (lines 1024–1043).

Reviewer's comment #4:

- Power analysis. Although the sample size in this study looks good to me, it could be useful to perform a power analysis based on previous studies or the number of trials.

Authors' response:

In response to the reviewer's comment, we performed a power analysis, based on the current sample size ($N = 118$) rather than by estimating the appropriate sample size in a post hoc way. In particular, one critical statistical test in the current study was the language component of the left cortical activity during the incongruent trials in the Stroop task (lines 458–470). To test this effect, as noted in our response to comment #1 of reviewer #2, we first contrasted the neural effect of interference (incongruent vs. congruent trials) between the left and right IPFC for each of the Stroop and Swimmy tasks. Then, the between-hemisphere difference in the interference effect was contrasted between the Stroop vs. Swimmy tasks, and subjected to a two-sample t-test. This t-test entailed testing the three-way interaction of trial (incongruent, congruent), laterality (left, right), and tasks (Stroop, Swimmy).

We observed a significant effect [$t(116) = 4.9, P < .001$], and the effect size was sufficiently large (Cohen's $d: 0.91$) (Fig. 7c; lines 463–465). Then, given the t -value, effect size, sample size, and alpha-rate (.001), we estimated the statistical power, which was high (0.94). We now describe this point in the Results (lines 465–466).

Reviewer #2

Reviewer's comment #1:

1. First of all, although prefrontal regions are consistently found in the Stroop task (incongruent vs. congruent conditions). Whether the activation was lateralized seems to be inconsistent (just to name a few: Bilateral: Leung, et al., 2002; Spielberg, et al., 2011; Peterson, et al. 2002. Right-lateralized: Bench, et al., 1993). This is critical to the current study: Why were lateralized results found in some studies but not the others?

Authors' response:

We appreciate the reviewer's crucial point about the hemispheric laterality. Lateralization of activity, by definition, indicates that the activity is greater in a region in one hemisphere than in the contralateral region. Neuroimaging studies of the Stroop effect, including those listed above by the reviewer, reported statistically significant activations but the activation laterality was inconsistent; some identified right hemisphere regions (e.g., Bench et al., 1993; Schulte et al., 2009; Egner, 2011), while others identified left hemisphere regions (e.g., Taylor et al., 1997; McKeown et al., 1998; Zysset et al., 2001; Galer et al., 2015; Freund et al., 2021) or bilateral regions (e.g., Banich et al., 2000; Leung et al., 2000; Egner and Hirsch, 2005; Spielberg et al., 2011; Peterson et al., 2002). This situation is reflected in the meta-analysis of the Stroop effect (Yarkoni et al., 2008; <https://neurosynth.org/analyses/terms/stroop/>).

The argument for hemispheric laterality based on indirect comparisons between significantly activated regions in previous studies is not strong because significant regions can change depending on the statistical threshold. For example, it is possible that a supposedly inactive hemisphere exhibits activity just below the statistical threshold. It is also possible that both hemispheres show significant activity, but one shows greater activity than the other, as in the current study (Figs. 3a and 4a and Supplementary Tables 1 and 2). As these possibilities illustrate, the arbitrariness of the statistical threshold for significant activity and the absence of testing hemispheric differences in activity are major reasons that variations of lateralization were found in previous studies.

To circumvent this issue, the current study, like our previous study (Tsumura et al., 2021), directly contrasted activity between the right and left hemispheres and explored brain regions showing greater activity in one hemisphere than in the other on a voxel-by-voxel basis (Supplementary Fig. 4; lines 973–986). We note that in this analysis, we explored only brain regions showing positive activity ($P > .05$ uncorrected, lines 982–983) to exclude biases due to the possible differential polarity of activation across hemispheres. We have clarified these points in the Discussion (lines 571–581).

Moreover, to confirm the robustness of hemispheric laterality, we performed a new ROI analysis in which cognitive control regions identified by Neurosynth (<https://neurosynth.org/analyses/terms/cognitive%20control/>) were divided into left lateral, medial, and right lateral regions. The results are now shown in a new figure (Fig. 7c). In the Stroop task, both left and right lateral regions showed significant activity [left: $t(62) = 11.1$, $P < .001$; right: $t(62) = 6.3$, $P < .001$], but the activity in the left hemisphere was greater than that in the right hemisphere [$t(62) = 8.41$, $P < .001$]. In the Swimmy task, both hemispheres showed significant activation [left:

$t(54) = 7.8, P < .001$; right: $t(54) = 7.6, P < .001$], and the activity did not differ between hemispheres [left vs. right: $t(54) = 1.44, P = .15$]. We now describe this result in the Results (lines 458–463).

We also compared the IPFC activity within the ROIs in the left and right hemispheres. As shown in the original manuscript, activity during the Stroop task was greater in the left hemisphere than in the right hemisphere [$t(62) = 6.8, P < .001$], but in the Swimmy task, there was no difference between hemispheres [$t(54) = 0.66, P = .51$] (lines 484–492; Supplementary Fig. 18b).

As these results demonstrate, we believe that direct comparisons of activity between hemispheres provide strong evidence of laterality. Such direct comparisons have rarely been performed in studies of the Stroop effect. We hope that our analysis, results, and discussion satisfactorily address the reviewer's concern.

Reviewer's comment #1 (cont.):

Although the current study has attempted to use the “language component” to explain the lateralization, I am not sure if the attempt is successful. First, although the Stroop effect is based on verbal interference, how much the activation in the left prefrontal cortex can be attributed specifically to linguistic information remains elusive. For example, in a patient study, although Cipolotti et al. (2016) found that performance in a Stroop task relies heavily on the left lateralized frontal regions. Another verbal task, however, relies on the right frontal regions.

Authors' response:

We appreciate the reviewer's important point about the verblity of the left IPFC activity during the Stroop task. We wish to note that one previous study directly compared brain activity between the Stroop task and non-verbal Simon task (Peterson et al., 2002). It showed that the left IPFC region showed greater activity in the Stroop task than in the Simon task, consistent with our results directly comparing activity in the Stroop and Swimmy tasks (Fig. 6a). Our results extend the previous findings by demonstrating that the verblity effect was also observed in the cerebellum. We now describe this point in the Discussion (lines 614–617).

In response to the reviewer's comment, we examined the effect of stimulus verblity on IPFC activity in the left hemisphere.

In the original manuscript, we defined IPFC ROIs in the left and right hemispheres based on the cognitive control regions identified in Neurosynth (see also our response above), and activity magnitudes within ROIs for the incongruent vs. congruent trials were extracted for the Stroop and Swimmy tasks. The activity magnitude was first contrasted between ROIs in the left and right hemispheres. In the Stroop task, activity was greater in the left hemisphere than in the right hemisphere [$t(62) = 6.8, P < .001$]. Such differential activity between the hemispheres was not observed in the Swimmy task [left vs. right: $t(54) = 0.66, P = .51$] (Supplementary Fig. 18a; lines 484–492).

In the revised manuscript, we extended this ROI analysis to examine the stimulus verblity component in the left IPFC. Specifically, the hemispheric difference in activity was compared between the Stroop and Swimmy tasks. The Stroop effect showed a greater hemispheric difference than the Swimmy effect [$t(116) = 4.5, P < .001$]. Importantly, the effect size was relatively large (Cohen's $d: 0.84$) (Cohen et al., 1988; Plonsky and Oswald 2014) (lines 492–494).

Finally, we calculated this effect size based on all lateral cortical areas, including the PPC and OTC. The results are now shown in a new figure (Fig. 7c). We found that the hemispheric difference in IPFC activity was greater in the Stroop task than in the Swimmy task [$t(116) = 4.9, P < .001$], and as with the analysis above, the effect size was large (Cohen's $d: 0.91$). Crucially, because the effects were controlled in terms of stimulus verbality (Stroop vs. Swimmy) and laterality (left vs. right), these large effect sizes suggest that greater left IPFC and neocortical activity during the incongruent trials in the Stroop task reflect language functions. We now describe these results in the Results (lines 458–470).

As the reviewer pointed out, the neuropsychological study of the Stroop task by Cipolotti et al. (2016) is consistent with our study in terms of the role of the left IPFC in resolving the Stroop effect. Cipolotti and colleagues also found that damage to the right IPFC impaired performance in a neuropsychological test using language materials. This test, the Hayling Sentence Completion Test (HSCT), Section 2 (Burgess and Shallice, 1997), requires withdrawal of prepotent and initiated inappropriate responses, namely, response inhibition (Burgess and Shallice, 1997; Cipolotti et al., 2016). Previous neuropsychological and neuroimaging studies have consistently suggested that response inhibition involves the right IPFC (e.g., Aron et al. 2004; Tsumura et al. 2022). Thus, it is reasonable that the performance on the HSCT is impaired by damage to the right IPFC, even though the HSCT uses verbal materials.

On the other hand, although cognitive interference occurred in both the current Stroop and Swimmy tasks (i.e., color–word interference in the Stroop task and outline–individual interference in the Swimmy task), the prepotency of the response toward one stimulus modality may not be strong compared to the HSCT, go/no-go tasks, or standard response inhibition tasks.

We now describe these points in the Discussion (lines 601–605).

Reviewer's comment #2:

2. The key research question and findings are not new. Although I like how the authors target the language component in Stroop, previous MRI studies have compared Stroop with other interferences, which rely less on verbal information. For example, Ven Veen & Carter (2005) have attempted to separate response and semantic interferences in the Stroop task and found that different brain regions were involved in the two interferences. Also, the current study did not significantly deviate from the authors' previous study (Morimoto, et al. 2008) in which they showed that in a Stroop-based flanker task, word interference triggered left inferior frontal gyrus (IFG) activities while color interference triggered right IFG activities. In that study, the authors explained: "...These results suggest that the verbal/nonverbal hemispheric specialization in the IFG can be explained by cognitive control processes per se." I think more work is needed to explain the major advances offered by the current study over previous studies.

Authors' response:

We appreciate the reviewer's critical point about the novelty of our question and findings. We wish to note that the key question of the current study is unrelated to that in the study by van Veen and Carter (2005), which attempted to isolate the response interference effect from the semantic interference effect and then reported

that the IPFC is associated with response interference bilaterally.

Our previous study (Morimoto et al., 2008) found verbality-dependent hemispheric asymmetry of IPFC involvement in flanker-type interference, and the current study significantly extended the previous study, as described below.

The most critical question of the current study concerns the cerebellum. In particular, as in the title and the Introduction (lines 58–72), we asked whether the cerebellum plays an important role during Stroop effect resolution involving cognitive and language processing. More specifically, we asked whether a cerebro–cerebellar loop, which has been examined in anatomically and in terms of sensorimotor functions, is also involved in the Stroop effect. Despite its importance and generality, the role of the human cerebro–cerebellar loop in higher cognitive functions remains elusive. This study is the first demonstration of the engagement of this loop in the resolution of the Stroop effect, and possibly in cognitive control functions of any kind. Thus, in response to the reviewer’s comment, we have highlighted the key question about the cerebellum by creating a schematic illustration that is now shown in a new figure (Fig. 1b). Additionally, we now clearly state the question in the Introduction (lines 70–72).

We found that the resolution of the Stroop effect involved cerebellar regions in the lateral and dorsal parts of the cerebellar cortex (Fig. 3a, Supplementary Table 1; lines 186–190). The involvement was right lateralized, which contrasts with the left-lateralized involvements of neocortical regions (Fig. 3b, lines 233–237). This cross-hemispheric involvement was highly consistent with prior studies involving transneuronal tracers (Strick et al., 2009; lines 529–532), sensorimotor systems (Diedrichsen et al., 2019; lines 536–537), and resting-state functional connectivity (Buckner et al., 2011, 2013; lines 537–542).

Importantly, our directional functional connectivity analysis revealed that the cerebellar regions send inhibitory signals to the right IPFC, whereas the IPFC sends excitatory signals to the cerebellum (Fig. 3c; lines 250–254). This inhibitory–excitatory signaling between the neocortex and cerebellar cortex has been demonstrated in animal neurophysiological and anatomical studies (Strick et al., 2009; Diedrichsen et al., 2019), but has never been reported in human higher cognitive functions. Thus, in response to the reviewer’s comment, we have highlighted our novel finding by creating a new figure, shown in Fig. 8b, that illustrates the cross-hemispheric excitatory–inhibitory signaling. We have also rearranged Figs. 3c to highlight this cross-hemispheric signaling.

Moreover, we functionally characterized the cerebellar involvement in the Stroop effect by performing a new analysis in response to comment #1a of reviewer #3 (see also our response to that comment below). The results are now shown in a new figure (Fig. 5b). We found that the activation locus in crus I/II was located in a region implicated in language function, as identified by a functional parcellation study (King et al., 2019). On the other hand, the activation locus in crus I/lobule VI was located in a region implicated in attention (King et al., 2019). These results suggest that language and attention functions implemented in the cerebellar cortex play an important role in resolving the Stroop effect. We now describe these results in the Results (lines 381–409). We believe that our findings pertaining to language and attentional processing in the cerebellum provide important insights into the neural mechanisms underlying the resolution of the Stroop effect.

We hope the reviewer's concern is satisfactorily addressed by our clarifications, our highlights of our question and findings, and our additional analysis characterizing the cerebellar role in the Stroop effect.

Reviewer's comment #3:

3. In multiple places, the authors claimed to find how the Stroop effect is resolved in the brain (and of course where the interference occurs), including in the abstract. I am not sure if I entirely understand this attempt. By resolution, do the authors mean that some neural mechanisms can alleviate the Stroop effect? I think this is certainly not the goal given the experimental design and findings here. So, I guess the authors refer to where the Stroop occurs in the brain. If so, how is that any different from asking where the interference occurs? More explanation on this point is required.

Authors' response:

The resolution refers to the critical mental operation necessary to achieve a behavioral goal (i.e., make a correct response) in the incongruent trials, and not to the alleviation (reduction) of the Stroop effect that can occur through training and practice. Our results suggest that the resolution is completed prior to response generation (Figs. 5 and 8c).

In theory, the terms resolution and interference are used differently (Braver, 2012). Specifically, when we refer to interference, we mean a situation where the behavioral goal is provided and the visual stimulus is perceived, but the critical processing needed to achieve the goal is not yet complete. On the other hand, the resolution indicates achievement of the goal. In other words, the interference (Stroop effect) occurs first, followed by its resolution.

A key question in the current study is how the Stroop effect is resolved in the brain. This question concerns not only neural correlates (i.e., where in the brain), but also task-related signal processing (i.e., what signals in the brain). For the former, we performed an exploratory analysis across the brain and identified multiple brain regions, including the IPFC and cerebellum. For the latter, we performed a directional functional connectivity analysis based on DCM and found inhibitory–excitatory signaling between the IPFC and cerebellum. As such, the current study provided neurophysiological evidence to answer the question.

We clarified this point in the Discussion (lines 712–726). We hope our explanation and revision clarify the usage of the terms and phrases.

Reviewer's comment #4:

4. Because four separate groups were recruited and performed the Stroop x Swimmy/vocal x manual tasks, I would like to see the authors discuss the possibility of group-specific results and why they chose between-subject over within-subject design in the current study.

Authors' response:

We appreciate the reviewer's important point about our experimental design. Samples were independent across the four groups (vocal Stroop, manual Stroop, vocal Swimmy, and manual Swimmy), and the size of each sample group was comparable to those in standard fMRI studies. All four groups showed robust effects (Figs. 2 and 5 and Supplementary Figs. 2, 5–11, and 14–16). We

acknowledge that our results could contain group-specific effects due to the nature of between-group comparisons, but given the sample size and robust results of each group, the degree of these group-specific effects should be equivalent to those in standard fMRI studies.

We used a between-subject design rather than a within-subject design because we wanted to minimize the training effect on task performance, since this effect changes the degree of interference in the Stroop task (Stroop, 1935; MacLeod 1991, for a review). For example, in a within-subject design, when the manual Stroop task is preceded by the vocal Stroop task, the vocal Stroop task serves as a training session for the manual Stroop task regarding the perception of the color–word stimulus and the resolution of color–word interference. It is also possible that through the four experiments, generic performance in stimulus perception and response generation for incongruent trials could be trained. We understand that counterbalancing the order of the conditions minimizes contamination. However, we were concerned that training would make the interference resolution more efficient (i.e., so that it would not require high cognitive demand), which would weaken the total interference effect across the four conditions. Therefore, we used a between-subject design where the amount of training for the incongruent trials was equivalent across the four conditions.

We have clarified these points in the Discussion (lines 626–632) and Methods (lines 771–774).

Other comments:

Reviewer’s comment #1:

1. P.11. Lines 259. The authors state “...activity was not as strong in the mPFC/ACC.” I am not sure which comparison was made to reach this conclusion.

Authors’ response:

The reviewer is correct. Then, in response to the reviewer’s comment, we deleted the sentence (line 273).

Reviewer’s comment #2:

2. P.13. Lines 306-311. The comparison seems to be between vocal vs. manual. However, the authors reported greater activities in some regions (e.g., primary motor cortex) for both contrasts. That is odd. Please clarify this.

Author’s response:

The reviewer is correct that the comparisons were made between vocal and manual conditions. We apologize that the original manuscript did not clearly describe the anatomical locations of the activated regions in the primary motor cortex and cerebellum.

In the primary motor cortex, ventrolateral regions were activated in the vocal condition (arrowheads labeled “vocalization” in Supplementary Figs. 13a and b *top*) because vocalization involves muscle movements of the tongue and mouth, whereas dorsomedial regions were activated in the manual condition (arrowheads labeled “right hand” in Supplementary Figs. 13a and b *top*) because the manual response involves finger movements. In the cerebellum, dorsal regions in lobules V/VIIIa/VIIIb were activated in the vocal condition (arrowheads labeled “vocalization” in Supplementary Figs. 13a and b *bottom*), whereas ventral regions

of the cerebellar lobules V/VI/VIIIa/VIIIb were activated in the manual condition (arrowheads labeled “right hand” in Supplementary Figs. 13a and b *bottom*). We clarified these points in the Results (lines 321–329). We also added labels “vocalization” and “right hand” in the cerebellar activations in Supplementary Figs. 13a and b.

Reviewer’s comment #3:

3. P.14. Lines 321-324. The authors conclude that “These results suggest that the roles of the LPFC, PPC, OTC, and cerebellum in the interference effects are independent of response modalities in both the Stroop and Swimmy tasks, and the generation of a response is not critical for resolution of the interference.” I think this claim is too strong. Given the results, these regions appeared in both tasks regardless of the modality, suggesting that modalities did not affect the activities. This is very different from independence.

Authors’ response:

We appreciate the reviewer’s critical point about the inference. In response to the reviewer’s comment, we redrafted the sentence to indicate that the response modality did not affect activity in the neocortical association regions or dorsolateral cerebellar hemispheres (lines 339–341).

Reviewer’s comment #4:

4. Currently, the number of runs/trials completed for each participant is not described. This will be crucial for future replications.

Authors’ response:

All participants completed five runs, each of which consisted of 48 trials (12 trials × 4 conditions). We clarified this point in the Methods (lines 880–881), and also reported the numbers of correct responses (lines 881–890).

Reviewer’s comment #5:

5. P.27. Lines 671-678. I appreciate that the authors tried to minimize motion-derived artifacts from vocal responses. However, I am not sure I understand how it was achieved. This is critical because, as a reader, I would like to know how head motion caused by vocal responses drove the brain activities.

Authors’ response:

It is known that head motion produces fMRI image artifacts, including distortion, signal dropout, and spike-like signals in timecourses, and that these occasionally increase type I/II errors in imaging analyses (e.g., Power et al., 2012; Siegel et al., 2014; Pruim et al., 2015; Ciric et al., 2017). In the current study, the head motions were greater in the vocal conditions than in the manual conditions, as revealed in a supplementary analysis performed in response to comment #1c of reviewer #3. This raises the question of whether head motion–derived artifacts contaminated our data in the vocal conditions.

To minimize image artifacts, we used a regression approach recommended by a benchmarking study (Ciric et al., 2017). Specifically, we first performed a GLM estimation where motion parameters and MRI signal timecourses (cerebrospinal fluid, white matter, and whole brain), as well as their derivatives and quadratics, were coded as regressors. Then, these effects were regressed out from image data.

Finally, the residual of the image was used in the subsequent GLM analyses to estimate task-related activations (lines 917–925).

Additionally, in response to comment #1c of reviewer #3, we performed supplementary preprocessing to reduce motion-derived artifacts using ICA-based denoising (Pruim et al., 2015) and motion censoring (scrubbing) (Power et al., 2012; Siegel et al., 2014; Davis et al., 2017; Bukkour et al., 2017). These points are now described in the Methods (lines 917–944).

Reviewer #3

Reviewer's comment #1:

1) The methodological approach lacks rigor and specificity.

a. There are not many details on the registration, but if the authors used SPM normalization without segmentation the cerebellum results may be biased with respect to the cerebellum (Diedrichsen et al., 2009, 2011). From the Group-level statistics section on p. 28, it sounds like SUIT was used to generate figures, but the cerebellar analyses were not carried out in SUIT space. But even when SUIT is used, studies have shown that cerebellar functional parcellations don't necessarily follow lobular boundaries (Bernard et al. 2012 and Buckner et al., 2011, using resting state fMRI, and more recently King et al. 2019 using task based fmri). I would have more enthusiasm for the paper if the authors used SUIT and a functional parcellation e.g., King to define language and non-language posterior cerebellar ROIs.

Authors' response:

We greatly appreciate this constructive suggestion on the functional characterization of cerebellar involvement. Regarding the registration, the reviewer is correct that in the original manuscript, we registered to structural and functional images to a standard MNI template using SPM12, and used SUIT only to project statistical maps onto 3D surface of the cerebellum. We have clarified this point in the Methods (lines 970–971).

In accordance with the reviewer's suggestion, we carried out a new GLM analysis in the cerebellum using the SUIT space and functionally characterized activated regions by referring to functional parcellation (King et al., 2019). The results are shown in the new Fig. 6b and Supplementary Figs. 14–16 in the revised manuscript.

Specifically, we registered structural and functional cerebellar images to the SUIT template by the following procedures, which are described at <https://www.diedrichsenlab.org/imaging/suit.htm>. Functional images were realigned and then coregistered to each individual's high-resolution structural image. The individual's cerebellum and brain stem were then extracted from the image based on tissue segmentation and tissue probability calculations implemented in SUIT and SPM12. As previously recommended (https://www.diedrichsenlab.org/imaging/suit_function.htm), for each participant an author (KJ) confirmed that the cerebellum and brain stem were appropriately extracted.

Next, the extracted structural image was registered to the SUIT template using nonlinear deformation based on Dartel implemented in SUIT/SPM12. Then, based on the estimated transformation matrix and deformation field as well as the cerebellar mask, functional images were registered to the SUIT space and resampled into 1-mm isotropic voxels. Finally, the images were spatially smoothed using 6-mm FWHM Gaussian kernel, and motion-related denoising was performed, as in the whole-brain preprocessing based on the MNI space.

Single-level GLM analysis of the cerebellar images was then performed, as in the whole-brain analysis. Group-level analysis was performed similarly, but the FWE rate was corrected within the cerebellum. Then, the cerebellar regions showing significant activity for the contrast of the incongruent and congruent trials

in the Stroop or Swimmy tasks were identified ($P < .05$, FWE-corrected). For the regions showing significant activity in the Stroop or Swimmy tasks, we performed conjunction and disjunction analyses in which the cerebellar regions were classified into those showing activity 1) only in the Stroop task, 2) only in the Swimmy task, or 3) in both tasks, and created conjunction and disjunction images.

As pointed out by the reviewer and demonstrated by previous studies (Buckner et al., 2011; King et al., 2019), the anatomical boundary of the cerebellum does not fully fit functional parcellations. Therefore, in response to the reviewer's comment, we defined ROIs as those in a functional parcellation based on task-fMRI data of a multi-domain task battery (King et al., 2019) in which 10 ROIs were defined. The conjunction and disjunction images were masked by each ROI image and unfolded into 2D flat maps of the cerebellum.

As shown in a new figure (Fig. 6b), crus I and II of Region 9 (King et al., 2019) showed significant activity only in the Stroop task. Interestingly, King et al. (2019) labeled this ROI as “verbal fluency”, a language-related function, and showed that it was restricted to the right hemisphere. Notably, in the whole-brain analysis in the current study, this region also showed greater activity in the Stroop task than in the Swimmy task (Fig. 6a).

In Regions 5 and 6, both of which were previously labeled as “divided attention” (King et al., 2019), bilateral regions in crus I and lobule VI were active in both the tasks. In these ROIs, the two structures showed activity in the right hemisphere in the Stroop task and in both hemispheres in the Swimmy task.

We created statistical z-maps showing this contrast (Supplementary Fig. 14) (uncorrected, unthresholded). We also created z-maps demonstrating the contrast of the Stroop vs. Swimmy tasks (Supplementary Fig. 15); these maps correspond to Fig. 6a in the whole-brain analysis. These maps reflect well the conjunction and disjunction analyses within the ROIs above.

These results suggest that cerebellar involvement is associated with language-related functionality in the Stroop task and cognitive functionality in both the Stroop and Swimmy tasks.

We also calculated group-level statistics for the contrast of the incongruent vs. congruent trials in the vocal condition relative to the manual condition, which corresponds to the whole-brain analysis in Fig. 5. Supplementary Fig. 16 shows that in Region 2, which was labeled as “right-hand presses” (King et al., 2019), there was prominent activity in the manual condition, confirming the results of the whole-brain analysis.

We now describe the procedures for the additional analyses in the SUIT space and their results in the Results (lines 381–409), Discussion (lines 548–551), and Methods (lines 1119–1149).

Reviewer's comment #1 (cont.):

b. I was disappointed in the lack of spatial specificity in the paper with regards to the lateral PFC. The lateral PFC is large and diverse and there is no acknowledgement of this in the analyses. I find it hard to interpret the results when they are generalized to such a large region. ROIs were defined for Stroop based on Swimmy activation and vice versa, but I didn't see a clear rationale for this besides wanting to use independent data. I would have preferred to see a theoretically

informed ROI, perhaps using the neurosynth ROIs or using data-based parcellations of the frontal lobe (e.g., Neubert et al., 2013; Sallet et al., 2013; Glasser et al., 2016). Another approach that would have improved specificity is using a voxel-based approach for the group-level rather than cluster-based. See Woo et al., 2009 for an excellent discussion of the issues with cluster-based approaches.

Authors' response:

We appreciate the reviewer's important comment about the anatomical and functional specificity of the IPFC ROI. We agree that the lateral prefrontal cortex implements diverse functions, and thus the use of a large IPFC region as an ROI makes it hard to interpret the results of the DCM analysis.

In response to the reviewer's comment, we performed a new DCM analysis in which IPFC ROIs were defined based on a meta-analysis map of cognitive control (<https://neurosynth.org/analyses/terms/cognitive%20control/>; association test). Specifically, the ROIs were defined in the peak coordinate within Brodmann area (BA) 44 in the IPFC. BA 44 is known to implement multiple cognitive control functions involved in the multiple demand system (Camilleri et al., 2018) and to serve as a core prefrontal region for language functions (Sakai, 2005). The exact coordinates were (-46, 18, 30; $z = 5.6$) and (46, 18, 32; $z = 7.6$) in the left and right IPFC, respectively. The ROI images were then created as spheres with 6-mm radii centered at the coordinates. Using these IPFC ROIs and the cerebellar ROIs, we estimated the connectivity parameters of the DCM. The results are now shown in new figures (Supplementary Figs. 5 and 10). Our initial findings were replicated in this additional analysis. We now describe these points in the Results (lines 254–256, 305–308) and Methods (lines 1020–1023, 1048–1053).

We acknowledge that the original manuscript was unclear regarding the definition of IPFC ROIs. We have now clarified the definition, specification, and rationale of our original IPFC ROIs in the revised manuscript, as below.

Specifically, in the original analysis, we defined IPFC ROIs by identifying the peak activation coordinates in BA 44 for the contrast of the incongruent and congruent trials for the vocal Stroop, manual Stroop, vocal Swimmy, and manual Swimmy tasks. Because of the asymmetric hemispheric involvements (i.e., the right hemisphere is more prominent), only right-hemisphere ROIs were identified in the Stroop task. The exact coordinates were (-40, 14, 28; $z = 4.67$) in the vocal condition, and (-44, 18, 28; $z = 6.06$) in the manual condition. By contrast, for the Swimmy task, bilateral IPFC ROIs were identified because of the absence of hemispheric asymmetry. The exact coordinates were (left: -54, 12, 30; $z = 4.00$) and (right: 52, 12, 28; $z = 2.47$) in the vocal condition, and (left: -48, 8, 26; $z = 4.21$) and (right: 46, 8, 30; $z = 3.73$) in the manual condition. The ROI images were then created as spheres with 6-mm radii centered at the coordinates. It should be noted that these ROIs are included in the meta-analysis map of cognitive control in Neurosynth ($P < .01$, FDR-corrected, uniformity test, minimum z -value: 8.4).

Importantly, to avoid circular analysis (Kriegeskorte et al., 2009), the DCM analysis of the vocal Stroop task used the IPFC ROI defined by the manual Stroop task, and vice versa. Similarly, the DCM analysis of the vocal Swimmy task used the IPFC ROI defined by the manual Swimmy task, and vice versa. In terms of functional specificity, the interchange of ROIs between vocal and manual

conditions within the Swimmy and Stroop tasks is valid because the IPFC region did not show differential activity between the vocal and manual conditions.

We hope that our IPFC ROI definition is reasonable regarding regional validity, functional specificity, and statistical procedure.

We note that similar procedures were used to define the cerebellar ROIs. In the Stroop task, the exact coordinates were (crus I/lobule VI: 32, -62, -28; $z = 4.96$) and (crus II/lobule VIIb: 24, -72, -42; $z = 4.85$) in the vocal condition, and (crus I/lobule VI: 30, -66, -30; $z = 5.25$) and (crus II/lobule VIIb: 26, -74, -46; $z = 5.72$) in the manual condition. In the Swimmy task, the exact coordinates were (right crus I/lobule VI: 20, -68, -20; $z = 5.31$), (left crus I/lobule VI: -24, -60, -26; $z = 4.62$), (right crus II/lobule VIIb: 30, -76, -48; $z = 4.37$), and (left crus II/lobule VIIb: -28, -70, -48; $z = 3.53$) in the vocal condition, and (right crus I/lobule VI: 16, -70, -24; $z = 5.44$), (left crus I/lobule VI: -22, -62, -26; $z = 4.78$), (right crus II/lobule VIIb: 28, -74, -46; $z = 3.95$), and (left crus II/lobule VIIb: -28, -70, -46; $z = 3.80$) in the manual condition.

We have clarified these points in the Methods (lines 1024–1047). Taken together, we hope that the new DCM analysis and the clarification of the ROI definition satisfactorily address the reviewer's concern.

Reviewer's comment #1 (cont.):

c. I was surprised to see no advanced pre-processing given the advanced imaging acquisition parameters. Echo Planar Imaging is prone to geometric distortion and further, multiband imaging is prone to motion induced 'banding artifacts', with greater banding seen with higher multiband factors. A number of preprocessing approaches have been developed to deal with these issues such as distortion correction, automatic and semi-automatic ICA denoising (e.g., ICA-AROMA and ICA-FIX), and motion censoring. Although multiband factor 8 data was collected, none of these preprocessing methods were performed.

Authors' response:

We acknowledge that because we used vocal response and multiband accelerated imaging, it is important to perform image preprocessing to reduce head motion-related artifacts. Notably, a previous study benchmarked denoising techniques used to reduce motion-related artifacts and showed that a GLM approach was most effective (Ciric et al., 2017). We note that in the original manuscript, we adopted this GLM approach (lines 917–925).

Specifically, functional images were subjected to GLM estimations in which motion parameters and MRI signal time courses (cerebrospinal fluid, white matter, and whole brain), and their derivatives and quadratics were coded as nuisance regressors. The estimations were performed using *fsl_regfilt* implemented in the FSL suite (<http://fmrib.ox.ac.uk/fsl/>). Then, a residual 4D image of the nuisance GLM was used for standard GLM estimations to extract task-related activity. We used this preprocessing in a previous study of task-related functional connectivity (Keerativittayayut et al., 2018) and in another study in which participants drank juice rewards during scanning, which produced head movements (Tanaka et al., 2020). We used multiband accelerated imaging in both studies.

Nonetheless, as the reviewer pointed out, because we used multiband accelerated imaging that was particularly susceptible to head motion-induced

artifacts, we newly performed preprocessing using ICA-AROMA (Pruim et al., 2015), which was applied to our original preprocessed functional images.

Moreover, we censored head motion (Power et al., 2012; Siegel et al., 2014). We first calculated motion magnitudes as framewise displacement (FD) values defined by Power et al. (2012). FD values were 0.192 ± 0.085 (mean \pm SD) in vocal Stroop, 0.011 ± 0.021 in manual Stroop, 0.196 ± 0.070 in vocal Swimmy, and 0.103 ± 0.032 in manual Swimmy. When FD was thresholded by 0.9 (Siegel et al., 2014; Davis et al., 2017), the rates of images that exceeded the threshold were 1.2 ± 3.6 % in vocal Stroop, 0.06 ± 0.1 % in manual Stroop, 1.0 ± 2.4 % in vocal Swimmy, and 0.1 ± 0.4 % in manual Stroop. The numbers of participants with images exceeding the threshold by more than 5% were two in the vocal Stroop task, zero in the manual Stroop task, one in the vocal Swimmy task, and zero in the manual Swimmy task. These results clearly indicate greater FD in vocal conditions. However, even in the vocal conditions, the absolute magnitudes were small compared to those in prior studies (Power et al., 2012; Siegel et al., 2014).

Then, using a GLM approach (Davis et al., 2017; Bakkour et al., 2017), we applied motion censoring (scrubbing) to the images in which ICA-AROMA was applied. We created a volume-wise regressor encoding 1 for volumes exceeding FD threshold (0.9) and 0 for others. Then, this effect was regressed out using *fsl_regfilt*.

We now show the results of applying ICA-AROMA and motion censoring in new figures (Supplementary Figs. 18–20). These results are very similar to those in Figs. 3–5 and 6a showing the original results, and confirm the primary findings.

We now describe these points in the Results (lines 498–503) and Methods (lines 926–940). We hope our theoretical consideration, empirical evidence, and additional analysis reasonably address the reviewer’s concern.

Reviewer’s comment #1 (cont.):

Unfortunately, there is no mention of field maps or blip-up/down acquisitions being collected, so I am worried that distortion correction could not be applied. Fieldmap-less distortion correction is available in the fmriprep advanced preprocessing pipeline, but there are no systematic evaluations of its success in functional imaging to my knowledge.

Although we used GRE-based field maps for high-order shimming, which is automatically applied prior to functional imaging in our MRI scanner (Siemens Prisma), field maps and blip-up/down data for distortion correction were unavailable in the current study. Thus, in response to the reviewer’s suggestion, we applied the fieldmap-less susceptibility distortion correction (SDC) implemented in fMRIPrep (Esteban et al., 2018). Technically, the fieldmap-less SDC is based on SDCFlows (<https://www.nipreps.org/sdcflows/master/methods.html#fieldmap-less-approaches>) and is implemented in the Advanced Normalization Tools (ANTs; <http://stnava.github.io/ANTs/>).

When testing the fieldmap-less SDC, we visually inspected structural and functional images on each step. Despite our intensive theoretical and technical efforts involving this approach, we found that distortions were not corrected but were even occasionally enhanced. We note that the initial fMRIPrep settings (Esteban et al., 2018) do not apply the fieldmap-less SDC, the SDCFlows clearly

warns that it is experimental, and as the reviewer pointed out, it has not yet been systematically evaluated. Thus, we think that there remain theoretical and technical issues in this approach and its implementation.

Given this situation, even though we used a modern, standard fMRI scanner with low distortion (Siemens, Prisma), we acknowledge that susceptibility distortions were not sufficiently corrected. Accordingly, the accuracy of the spatial identification of the functional localization could be limited. We now describe this point in the Methods (lines 941–944).

Reviewer’s comment #2:

2) The between subjects design limits the power to compare the Stroop and Swimmy task results. Indeed, the authors didn’t perform any analyses that directly compared the groups, focusing instead on qualitative differences between results. Given the large sample size, I expect that the authors had adequate power to perform a group comparison, rather than use qualitative comparisons which may yield flawed inferences. If statistically significant results are not found in the group comparison, the authors could report effect size, which would be more informative for designing follow up studies than non-significant p-values.

Authors’ response:

We wish to note that in the original manuscript, we statistically performed group-level comparisons between the Stroop and Swimmy tasks (Fig. 5c in the original manuscript). Specifically, we performed a whole-brain exploratory analysis to identify brain regions showing differential activity between the tasks by directly contrasting the interference effects between them. In response to the reviewer’s comment, we have highlighted this analysis by moving the results to Fig. 6a from Fig. 5c. The full coordinates for significant regions are listed in Supplementary Table 5.

The interference effect was greater in the IPFC and PPC in the left hemisphere and in crus II in the right cerebellum. On the other hand, the Swimmy task showed a greater interference effect in the PPC in the right hemisphere (lines 359–367).

Reviewer’s comment #2 (cont.):

a. The abstract doesn’t mention the Swimmy task at all, raising the question of whether this task should be reported in this manuscript. However, I find the task to be quite interesting and I would prefer to see a more direct comparison as mentioned above. Relatedly, there isn’t much information on the Swimmy task - it seems like a combination of a Simon task and a Local-Global task so the authors should relate the task to these existing tasks as well as to fMRI studies highlighting their neural processes.

Authors’ response:

We appreciate the reviewer’s comment highlighting the Swimmy task. In response to the comment, we now explicitly state in the abstract that the Swimmy task served as a control task for the Stroop task in terms of stimulus verblivity (lines 8, 13–14).

As we noted in response to comment #2 immediately above, we performed a statistical analysis to directly compare the activity in the Stroop and Swimmy tasks, and the results are shown in Fig. 6a, Supplementary Table 5, and the Results (lines 359–367).

Additionally, in the Introduction, we now explain the Swimmy task by referring to the Simon task (Simon, 1969) and the local–global tasks (Fink et al., 1996) (lines 77–79).

We also relate our results to those in previous neuroimaging studies of the Simon task (Peterson et al., 2002; Cespon et al., 2020) and the local–global task (Fink et al. 1996). In particular, bilateral IPFC involvement in the Swimmy effect (incongruent vs. congruent) was consistent with previous studies of the Simon task (Peterson et al., 2002; Cespon et al., 2020). Notably, the previous neuroimaging study directly compared activity during the Stroop task and the Simon task and found greater activity in the left IPFC during the Stroop task (Peterson et al., 2002), consistent with our results directly comparing the Stroop and Swimmy tasks (Fig. 6a, Supplementary Table 5). The contrast of the outline vs. individual tasks of the Swimmy task showed multiple neocortical and cerebellar regions (Supplementary Fig. 9c), reflecting distinct attention to global and local shapes as revealed by a previous study of the local–global task (Fink et al., 1996).

We now describe these points in the Results (line 279) and Discussion (lines 613–617).

Minor issues:

Reviewer’s comment #1:

1. The authors state that data and code are available upon reasonable request. Given the wide availability of resources to enable open science (OSF, github, openneuro, etc), I find this data availability statement to be unacceptable. By placing code and data in repositories the authors both help to ensure that data and code are not lost and/or the knowledge of how to interpret the code is not lost. I encourage the authors to openly share their data and code.

Authors’ response:

We assure the reviewer that we will share our data and codes in a public repository when the manuscript is accepted for publication. This has been clearly stated in the Reporting Summary and in the text (lines 743–745).

Reviewer’s comment #2:

2. I noted a couple of typos. Crus is misspelled as crura in 2 places and Flanker is misspelled franker one time that I noted.

Author’s response:

We corrected the typos pertaining to the flanker task and crus. We then double checked that the usage of the terms was correct throughout the text, figures, and tables.

REVIEWER COMMENTS

Reviewer #1 (Remarks to the Author):

The authors considered each point I raised in my first review, and some of them were addressed appropriately. However, I still disagree with some details.

About the asymmetry of the Stroop effects, the common outcome in the literature is a congruency effect in color naming but no congruency effect in word reading (i.e., no reverse Stroop effect), and not only a stronger Stroop effect (incongruent minus congruent) in color naming than in word reading, as the authors mentioned in the discussion of the manuscript. Here, a reverse Stroop effect is observed for vocal and manual Stroop tasks, with RT and accuracy, which is unusual in normal reading conditions, especially in the vocal task.

The authors suggest that a reverse Stroop effect can be observed when baseline cognitive demand is high, but studies reported to support this hypothesis do not seem appropriate to me (Dunbar & MacLeod, 1984; Durgin, 2000, 2003; Melara & Mounts, 1993). Furthermore, interpretations presented in the cited papers do not mention cognitive demand. In these four studies, participants do not need to determine first, on each trial, whether they have to process the color or the verbal information (depending on the direction of the vertex). Each task (color or word identification) is performed separately. In Durgin's study (2003), the task is pretty simple and relatively close from the classic (reverse) Stroop task. A color word was printed at the center of the screen and color patches were displayed around the word. Participants had to use the mouse cursor to point to the color patch corresponding to the color name, ignoring the printed color (or simply indicate whether the target color was present among the color patches; Experiment 2). In Dunbar and MacLeod (1984), a reverse Stroop effect is observed only when the readability of words is strongly degraded (as in Melara & Mounts, 1993), which is very different from the current study. Maybe this part should be modified or removed, but it is important to mention (as the authors did in the revised version of the manuscript, lines 675-676) that "the baseline cognitive demand was cancelled out when comparing behavior and neuroimaging data between trials and tasks."

I think reactive control is more convincing to explain the atypicality of the behavioral results. This interpretation should be developed more clearly in the manuscript. It is unusual to mix the two tasks (color naming and word reading) as the authors did in this study, which of course does not mean that the design is invalid. In this situation, proactive control is absent and only reactive control seems possible, as the authors noted. The specificity of this situation could explain, at least partially, the long RTs and, more importantly, the presence of reverse Stroop effects. This situation could both produce an increased processing of the irrelevant dimension and/or reduce the inhibition of the irrelevant

information, relative to a situation in which proactive control is present. As a consequence, reverse Stroop effects would be more susceptible to occur.

I wonder if the authors are aware about other Stroop studies in which color naming and word reading are intermixed as in the present study? If so, it would be interesting to discuss results of these potential studies in the manuscript.

The authors indicated in the discussion (lines 698-700) that “this task situation may have modulated the predominance of processing word and color, as demonstrated in previous studies (Dunbar & MacLeod, 1984; Melara & Mounts, 1993).” Both cited studies manipulated the discriminability of the two dimensions, which is not the case in the present study. Although I understand the rationale of the authors, I do not think these papers are very well appropriate to explain results observed in this study. They then mentioned that “this interpretation fits well with a theoretical account of the Stroop effect formulated and revised by previous studies (Cohen et al., 1990; Grégoire et al., 2014).” In the Cohen et al.’s connectionist model, the main variable is the strength of processing pathways, which is itself a function of the amount of training. Grégoire et al. refined this model by suggesting that the pattern of interference in the Stroop task depends on the relative strength of the two competing pathways (e.g., word reading and color naming). Thus, Grégoire et al. proposed that a similar degree of practice of the two competing processes in the Stroop task should produce a Stroop effect and a reverse Stroop effect of comparable magnitude. Insofar as the present study did not manipulate the practice of the two competing processes (and so, word reading is very likely much more practiced in most of participants), this model does not seem appropriate to explain results. Furthermore, what the authors called “processing predominance” looks somewhat confusing.

To summarize, I suggest modifying the two paragraphs (Stroop and reverse Stroop effects and Proactive and reactive control in Stroop tasks) in the Discussion (pp. 25-27). The authors should develop in more details what they have already mentioned about proactive and reactive control. Inappropriate references could also be removed to lighten the presentation.

Minor points

- I am not sure I understand what the authors mean by “language functions” (lines 678-679). Please clarify.
- The paragraph (lines 590-594) could be rephrased by toning down the language. For example: “We acknowledge that our contrast of the incongruent vs. congruent trials could involve a facilitation effect (faster response in the congruent trials than in the neutral trials). However, facilitation is not a concomitant of interference, and the facilitation effect is usually weaker than the interference effect. Thus, we do not think that the facilitation is dominant against the interference in this study.” Note that Kalanthroff et al. (2019, Psychol. Rev.) proposed that a low proactive control could lead to a reverse facilitation (i.e., faster responses to neutral stimuli than to congruent stimuli). Roelofs (2010, J. Exp.

Psychol. Learn. Mem. Cogn.) also suggested that Stroop interference and facilitation have a common locus.

Reviewer #2 (Remarks to the Author):

I appreciate the detailed responses from the authors. The manuscript is in a better shape now. My major worry remains the same, which is that the results were inconsistent with the literature in a few places. This was also noted by another reviewer. I understand and appreciate that the authors have pointed out potential factors contributing to these inconsistencies (e.g., the analysis (regarding whether bilateral activation is common in Stroop studies) and the design (concerning the differences in inhibitory process between the block and event-related design)). However, since these differences were not targeted and tested a priori, it is almost impossible to know whether they are “produced” by some artifacts in the current study, or genuinely reflect differences in different Stroop paradigms. That being said, after the revision, I can see the merits of the study more clearly, which is the role of cerebral-cerebellar connections in the Stroop task.

Nevertheless, I am still not convinced that the crosstalk between the frontal regions and the cerebellum reflects specifically the resolution but is not involved in the interferences. In response to my original comment 3, the authors differentiated interference and resolution conceptually. This I understand, I just don't think the current design is able to tease these two factors apart. Theoretically, the resolution of a Stroop effect happens when the agent “resolves” the conflict and makes a response (either implicitly or explicitly). However, when one measures this mental process, how do we know that the results reflect purely the resolution, the conflict (interference), or the combination of the two? This is key to the current study's main claim, and I look forward to seeing the authors' new argument/analysis/etc to support this claim.

Shao-Min (Sean) Hung

Reviewer #3 (Remarks to the Author):

The authors have done a thorough job responding to reviewer concerns. I no longer have any remaining concerns with the manuscript.

Response to Reviewer Comments

Manuscript by Okayasu et al. (manuscript number: NCOMMS-22-06757B)

Title: An excitatory–inhibitory fronto–cerebellar loop resolves the Stroop effect.

We wish to thank the editors and reviewers for handling and evaluating our revised manuscript. We were gratified to see the positive reactions to our manuscript. The reviewers expressed remaining concerns and provided constructive suggestions regarding the revised manuscript. In accordance with the reviewers' comments, we revised the Discussion as follows:

- 1) In response to the comments of reviewer #1, we described the cognitive demands imposed by our task in greater detail in the Discussion. We also removed irrelevant sentences and references.
- 2) In response to the comments of reviewer #2, we clarified that our study design does not make it possible to differentiate interference and its resolution, and this is not the aim of our study.

In this document, we present point-by-point responses to the reviewers' comments and indicate where we have made relevant revisions in the manuscript. Revisions are highlighted in red in the revised manuscript.

We hope that these revisions satisfactorily address the reviewers' concerns. The page numbers refer to the revised manuscript unless otherwise mentioned.

Reviewer #1

Reviewer's comment #1:

About the asymmetry of the Stroop effects, the common outcome in the literature is a congruency effect in color naming but no congruency effect in word reading (i.e., no reverse Stroop effect), and not only a stronger Stroop effect (incongruent minus congruent) in color naming than in word reading, as the authors mentioned in the discussion of the manuscript. Here, a reverse Stroop effect is observed for vocal and manual Stroop tasks, with RT and accuracy, which is unusual in normal reading conditions, especially in the vocal task.

The authors suggest that a reverse Stroop effect can be observed when baseline cognitive demand is high, but studies reported to support this hypothesis do not seem appropriate to me (Dunbar & MacLeod, 1984; Durgin, 2000, 2003; Melara & Mounts, 1993). Furthermore, interpretations presented in the cited papers do not mention cognitive demand. In these four studies, participants do not need to determine first, on each trial, whether they have to process the color or the verbal information (depending on the direction of the vertex). Each task (color or word identification) is performed separately. In Durgin's study (2003), the task is pretty simple and relatively close from the classic (reverse) Stroop task. A color word was printed at the center of the screen and color patches were displayed around the word. Participants had to use the mouse cursor to point to the color patch corresponding to the color name, ignoring the printed color (or simply indicate whether the target color was present among the color patches; Experiment 2). In Dunbar and MacLeod (1984), a reverse Stroop effect is observed only when the readability of words is strongly degraded (as in Melara & Mounts, 1993), which is very different from the current study. Maybe this part should be modified or removed, but it is important to mention (as the authors did in the revised version of the manuscript, lines 675-676) that "the baseline cognitive demand was cancelled out when comparing behavior and neuroimaging data between trials and tasks."

Authors' response:

We appreciate the reviewer's crucial point about baseline cognitive demand. We agree with the reviewer's point that these previous studies (Dunbar & MacLeod, 1984; Durgin, 2000, 2003; Melara & Mounts, 1993) did not impose high extra cognitive demand. In the previous manuscript, we used the term "cognitive" in a broad sense, and it encompassed stimulus perception and attentional shift. But we now understand that these previous studies are irrelevant to our study with regard to baseline cognitive demand. Accordingly, we deleted the sentences referring to these studies (lines 672, 674). We kept the statement that baseline cognitive demand is canceled out when comparing task conditions in the behavioral and imaging analyses, because the incongruent and congruent trials and the color and word tasks were pseudorandomized in the tasks (lines 674–677).

Reviewer's comment #1 (cont.):

I think reactive control is more convincing to explain the atypicality of the behavioral results. This interpretation should be developed more clearly in the manuscript. It is unusual to mix the two tasks (color naming and word reading) as the authors did in this study, which of course does not mean that the design is

invalid. In this situation, proactive control is absent and only reactive control seems possible, as the authors noted. The specificity of this situation could explain, at least partially, the long RTs and, more importantly, the presence of reverse Stroop effects. This situation could both produce an increased processing of the irrelevant dimension and/or reduce the inhibition of the irrelevant information, relative to a situation in which proactive control is present. As a consequence, reverse Stroop effects would be more susceptible to occur.

Authors' response:

We appreciate the reviewer's constructive and insightful comment about the interpretation of the robust reverse Stroop effect observed in the current study. We agree that the reactive control situation in our task could have yielded the reverse Stroop effect. The longer RTs are also attributable to the reactive control situation. We clarified these points in the Discussion (lines 697–700).

Reviewer's comment #1 (cont.):

I wonder if the authors are aware about other Stroop studies in which color naming and word reading are intermixed as in the present study? If so, it would be interesting to discuss results of these potential studies in the manuscript.

Authors' response:

We are not aware of previous Stroop studies using the intermixing trial design. As we indicated in the previous revision, we failed to see a robust Stroop effect in our pilot functional MRI experiment that used a blocked design in which trial frequency and eye movements were controlled, thus enabling strong proactive control. We were unable to establish a blocked design with a weak proactive control. We therefore designed a task where the congruent and incongruent trials were intermixed.

Reviewer's comment #1 (cont.):

The authors indicated in the discussion (lines 698-700) that "this task situation may have modulated the predominance of processing word and color, as demonstrated in previous studies (Dunbar & MacLeod, 1984; Melara & Mounts, 1993)." Both cited studies manipulated the discriminability of the two dimensions, which is not the case in the present study. Although I understand the rationale of the authors, I do not think these papers are very well appropriate to explain results observed in this study. They then mentioned that "this interpretation fits well with a theoretical account of the Stroop effect formulated and revised by previous studies (Cohen et al., 1990; Grégoire et al., 2014)." In the Cohen et al.'s connectionist model, the main variable is the strength of processing pathways, which is itself a function of the amount of training. Grégoire et al. refined this model by suggesting that the pattern of interference in the Stroop task depends on the relative strength of the two competing pathways (e.g., word reading and color naming). Thus, Grégoire et al. proposed that a similar degree of practice of the two competing processes in the Stroop task should produce a Stroop effect and a reverse Stroop effect of comparable magnitude. Insofar as the present study did not manipulate the practice of the two competing processes (and so, word reading is very likely much more practiced in most of participants), this model does not seem appropriate to explain results. Furthermore, what the authors called "processing predominance"

looks somewhat confusing.

Authors' response:

We appreciate the reviewer's significant insight about the computational theory underlying the Stroop effect. We now understand that the indicated sentences are irrelevant to our study, and we deleted them (line 701).

Reviewer's comment #1 (cont.):

To summarize, I suggest modifying the two paragraphs (Stroop and reverse Stroop effects and Proactive and reactive control in Stroop tasks) in the Discussion (pp. 25-27). The authors should develop in more details what they have already mentioned about proactive and reactive control. Inappropriate references could also be removed to lighten the presentation.

Authors' response:

As suggested by the reviewer, we specified the reactive nature of our task and its consequences in greater detail. Specifically, we indicated that the reactive control may have produced a robust reverse Stroop effect due to increased processing of the irrelevant dimension and/or reduced inhibition of the irrelevant information (lines 696–700). Moreover, we highlighted that the high baseline cognitive demand derived from the reactive control was canceled out in the analyses (lines 672–677). Finally, we removed irrelevant descriptions and references to clarify our points in the two paragraphs (lines 661–707). Given these revisions, we believe that the two paragraphs now better highlight the reactive control as a reasonable interpretation of the robust reverse Stroop effect without contaminating our results. We greatly appreciate the reviewer's crucial insights and suggestions and hope that our revisions satisfactorily address the reviewer's concerns.

Minor points:

Reviewer's comment #1:

- I am not sure I understand what the authors mean by "language functions" (lines 678-679). Please clarify.

Authors' response:

Here we are referring to visual word perception and semantic processing. In response to the reviewer's comment, we revised the sentence (line 671).

Reviewer's comment #2:

- The paragraph (lines 590-594) could be rephrased by toning down the language. For example: "We acknowledge that our contrast of the incongruent vs. congruent trials could involve a facilitation effect (faster response in the congruent trials than in the neutral trials). However, facilitation is not a concomitant of interference, and the facilitation effect is usually weaker than the interference effect. Thus, we do not think that the facilitation is dominant against the interference in this study."

Authors' response:

We appreciate the reviewer's suggestion. Accordingly, we rephrased these sentences (lines 591, 594, 598).

Reviewer's comment #2 (cont.):

Note that Kalanthroff et al. (2019, Psychol. Rev.) proposed that a low proactive

control could lead to a reverse facilitation (i.e., faster responses to neutral stimuli than to congruent stimuli). Roelofs (2010, J. Exp. Psychol. Learn. Mem. Cogn.) also suggested that Stroop interference and facilitation have a common locus.

Authors' response:

We appreciate the constructive suggestion. We incorporated these studies in the revised manuscript and fleshed out the description of facilitation and interference (lines 594–597).

Reviewer #2

Reviewer's comment #1:

I appreciate the detailed responses from the authors. The manuscript is in a better shape now. My major worry remains the same, which is that the results were inconsistent with the literature in a few places. This was also noted by another reviewer. I understand and appreciate that the authors have pointed out potential factors contributing to these inconsistencies (e.g., the analysis (regarding whether bilateral activation is common in Stroop studies) and the design (concerning the differences in inhibitory process between the block and event-related design)). However, since these differences were not targeted and tested a priori, it is almost impossible to know whether they are “produced” by some artifacts in the current study, or genuinely reflect differences in different Stroop paradigms. That being said, after the revision, I can see the merits of the study more clearly, which is the role of cerebral-cerebellar connections in the Stroop task.

Authors' response:

We appreciate the reviewer's positive reaction to our responses, and the recognition that a merit of our study is its demonstration of the involvement of a cerebro-cerebellar loop in the Stroop effect. We also appreciate the reviewer's important point about the differences between our study and previous studies regarding study design and results. The reviewer is correct that these differences in the analysis (i.e., whether bilateral activation is common in Stroop studies) and design (i.e., in terms of the inhibitory process between the block and event-related designs) were not targeted or tested a priori. In the current study, the lateralized activity was instead found by an exploratory analysis. We now describe this point in the Discussion (lines 576–577).

Notably, our task design was optimized for a standard functional MRI study that reduced potential confounds of oddball effects and proactive control (lines 680–690; 807–811; 820–836). Importantly, these effects were canceled out in all our analyses (lines 674–677). We added these two points in the first revised manuscript in response to comments #1 and #2 of reviewer #1. Given these aspects of the task design and analysis procedure, we do not think our results are produced only by artifacts derived from our task design.

In this revision, we agree with comment #1 of reviewer #1 stating that “reactive control is more convincing to explain the atypicality of the behavioral results.” Accordingly, we clarified in the Discussion that the reactive control situation in our task could have yielded the reverse Stroop effect, and that the longer RTs are also attributable to the reactive control situation (lines 697–700). We believe that this revision in the Discussion satisfactorily addresses the reviewer's concern.

Reviewer's comment #1 (cont.):

Nevertheless, I am still not convinced that the crosstalk between the frontal regions and the cerebellum reflects specifically the resolution but is not involved in the interferences. In response to my original comment 3, the authors differentiated interference and resolution conceptually. This I understand, I just don't think the current design is able to tease these two factors apart. Theoretically, the resolution of a Stroop effect happens when the agent “resolves” the conflict and makes a

response (either implicitly or explicitly). However, when one measures this mental process, how do we know that the results reflect purely the resolution, the conflict (interference), or the combination of the two? This is key to the current study's main claim, and I look forward to seeing the authors' new argument/analysis/etc to support this claim.

Authors' response:

We appreciate the reviewer's important point about the distinction between the interference and its resolution that we described in the previous revised manuscript. We agree that our study design did not allow us to differentiate these processes. Importantly however, this was not the intent of our study. Rather, we aimed to differentiate interference/resolution from response generation. To highlight our aim, in Fig. 1g we illustrate our study design, which consists of a 2 x 2 factorial design with two levels of stimulus modalities (Stroop and Swimmy) and response modalities (vocal and manual). Furthermore, to highlight the interpretation of our results, in Fig. 8c we present schematic processing diagrams of the Stroop task based on our results, as well as physiological and anatomical evidence. In these diagrams, we illustrate the two stages shown in Fig. 1g: interference resolution and response generation. However, these figures (Figs. 1g and 8c) do not differentiate resolution and interference.

Distinguishing active resolution from interference may require a control condition where Stroop-type interference occurs unconsciously, as demonstrated in a recent study (Hung et al. 2020).

We added these points in the Discussion (lines 715–719). We also clarified that our key question includes how the interference occurs and is resolved in the brain (i.e., not how to discriminate between the interference and resolution) (line 720). We hope our response satisfactorily addresses the reviewer's concern.

Reviewer #3

Reviewer's comment #1:

The authors have done a thorough job responding to reviewer concerns. I no longer have any remaining concerns with the manuscript.

Authors' response:

We greatly appreciate the reviewer's crucial, constructive, and insightful comments, which significantly improved our manuscript.

REVIEWERS' COMMENTS

Reviewer #1 (Remarks to the Author):

All my points were properly addressed. I don't have any further comments.

Reviewer #2 (Remarks to the Author):

The revisions have made the manuscript clearer. In general, I am happy to accept the manuscript as is.

However, I still think that my previous major comment was not addressed, and I would like to elaborate on it once again.

In the comment where I argued that the current design did not allow distinction between resolution and interference in the Stroop paradigm. The authors replied:

"We agree that our study design did not allow us to differentiate these processes. Importantly however, this was not the intent of our study. Rather, we aimed to differentiate interference/resolution from response generation."

I do not entirely follow the authors here. I guess that the authors were trying to say that since a correct response was made, the Stroop interference had to be resolved. I entirely agree. The question is the differentiation between interference and resolution. Since the entire process, from perception to response, happened in close temporal proximity, it was difficult for MRI to disentangle different sub-processes within this window. Based on the reply from the authors, I think they agree on this point. However, the central claim of the study is how the frontal-cerebellar loop "resolves" the Stroop effect while such brain activities could at least partially reflect interference. The authors cited Hung et al (2020) and suggested that an unconscious Stroop interference could allow differentiation of resolution and interference. I do not understand how though since the interference and resolution will still happen concurrently.

Response to Reviewer Comments

Manuscript by Okayasu et al. (manuscript number: NCOMMS-22-06757C)

Title: The Stroop effect involves an excitatory–inhibitory fronto–cerebellar loop.

We wish to thank the editors and reviewers for handling and evaluating our revised manuscript. We were gratified to see the positive reactions to our manuscript. The reviewers expressed remaining concerns and provided constructive suggestions regarding the revised manuscript. In accordance with the reviewers' comments, we changed the title and revised the Abstract and Discussion.

In this document, we present point-by-point responses to the reviewers' comments and indicate where we have made relevant revisions in the manuscript. Revisions are highlighted in red in the revised manuscript.

We hope that these revisions satisfactorily address the reviewers' concerns. The page numbers refer to the revised manuscript unless otherwise mentioned.

Reviewer #1

Reviewer's comment #1:

All my points were properly addressed. I don't have any further comments.

Authors' response:

We greatly appreciate the reviewer's crucial and constructive comments, which significantly improved our manuscript.

Reviewer #2

Reviewer's comment #1:

In the comment where I argued that the current design did not allow distinction between resolution and interference in the Stroop paradigm. The authors replied: "We agree that our study design did not allow us to differentiate these processes. Importantly however, this was not the intent of our study. Rather, we aimed to differentiate interference/resolution from response generation." I do not entirely follow the authors here. I guess that the authors were trying to say that since a correct response was made, the Stroop interference had to be resolved. I entirely agree.

Authors' response:

Yes, in the previous manuscripts, we stated that the Stroop interference was resolved because we analyzed correct trials.

Reviewer's comment #1 (cont.):

The question is the differentiation between interference and resolution. Since the entire process, from perception to response, happened in close temporal proximity, it was difficult for MRI to disentangle different sub-processes within this window. Based on the reply from the authors, I think they agree on this point.

Authors' response:

We agree that it is unable to dissociate interference and its resolution in the Stroop tasks based on the temporal resolution of the fMRI signal. We clarified this point in the Discussion (line 618).

Reviewer’s comment #1 (cont.):

However, the central claim of the study is how the frontal-cerebellar loop “resolves” the Stroop effect while such brain activities could at least partially reflect interference.

Authors’ response:

As we responded in the 2nd revision (NCOMMS-22-06757B), we clarified that our study design did not allow us to differentiate the interference and its resolution, which is now acknowledged by the reviewer.

In response to the reviewer’s further concern, in this revised manuscript, we changed the title of our manuscript to “the Stroop effect involves an excitatory-inhibitory front–cerebellar loop” which does not claim the resolution. Moreover, we revised the Abstract such that it avoids claiming the dissociation of interference and resolution based on our empirical evidence (lines 7, 10, 14-16). We hope that the change of the title and revision of the Abstract satisfactorily addresses the reviewer’s concern.

Reviewer’s comment #1 (cont.):

The authors cited Hung et al (2020) and suggested that an unconscious Stroop interference could allow differentiation of resolution and interference. I do not understand how though since the interference and resolution will still happen concurrently.

Authors’ response:

In response to the reviewer’s comment, we deleted the sentence (line 620).

We greatly appreciate the reviewer’s helpful and crucial comments to improve our manuscript.